# BitMark: Watermarking Bitwise Autoregressive Image Generative Models

**Louis Kerner, Michel Meintz, Bihe Zhao, Franziska Boenisch, Adam Dziedzic**
{louis.kerner, michel.meintz, bihe.zhao, boenisch, adam.dziedzic}@cispa.de
CISPA Helmholtz Center for Information Security

## Abstract

State-of-the-art text-to-image models generate photorealistic images at an unprecedented speed. This work focuses on models that operate in a bitwise autoregressive manner over a discrete set of tokens that is practically infinite in size. However, their impressive generative power comes with a growing risk: as their outputs increasingly populate the Internet, they are likely to be scraped and reused as training data—potentially by the very same models. This phenomenon has been shown to lead to model collapse, where repeated training on generated content, especially from the models' own previous versions, causes a gradual degradation in performance. A promising mitigation strategy is watermarking, which embeds human-imperceptible yet detectable signals into generated images—enabling the identification of generated content. In this work, we introduce BitMark, a robust bitwise watermarking framework. Our method embeds a watermark directly at the bit level of the token stream during the image generation process. Our bitwise watermark subtly influences the bits to preserve visual fidelity and generation speed while remaining robust against a spectrum of removal techniques. Furthermore, it exhibits high radioactivity, i.e., when watermarked generated images are used to train another image generative model, this second model's outputs will also carry the watermark. The radioactive traces remain detectable even when only *fine-tuning* diffusion or image autoregressive models on images watermarked with our BitMark. Overall, our approach provides a principled step toward preventing model collapse in image generative models by enabling reliable detection of generated outputs. The code is available at https://github.com/sprintml/BitMark.

## 1 Introduction

The image generation capabilities of large scale models [8, 11, 37, 12, 23, 26] have improved tremendously. These models can now generate at an unprecedented speed high-quality photorealistic outputs, which are often indistinguishable from real images [11, 37]. Several models [11, 37] now achieve state-of-the-art performance by operating in a *bitwise* autoregressive manner [35, 41]. This provides them with a near infinite number of possible tokens, yielding high-resolution images. This performance, however, comes at the expense of a training process that is highly time-, cost- and most of all data-intensive [25, 29]. The required training data are often scraped from the Internet [13, 24, 32]. This practice carries risks: with the ever-growing amount of generated data on the internet, the models get increasingly trained on *generated* data rather than *natural* data. The iterative training of models on their own generated data has been shown to degrade performance [1, 33], which is referred to as *model collapse*. The process also leads to encoding the current models' mistakes, biases, and unfairnesses into its future versions [40]. From the perspective of model owners, especially big companies such as OpenAI, Google, or ByteDance, who invest in large scale proprietary models that are exposed via public APIs, this poses a significant threat. To prevent the possible degradation of model quality, it is crucial for them to identify *their own* generated content.

39th Conference on Neural Information Processing Systems (NeurIPS 2025).

A promising way to address the above problems is watermarking [4, 9, 10, 14, 22, 38, 48]. By embedding human-imperceptible, yet detectable signals into generated images, a model owner can automatically identify and filter out their model's previously generated content when collecting training data for future versions. To prevent model collapse, watermarks should be radioactive [7, 28, 31]—that is, they should persist across model lineages, so that models trained on watermarked outputs also reproduce the watermark. This is crucial when third parties fine-tune models on such outputs and use them to generate new content, which may later be scraped and reused by the original model—causing it to (unknowingly) train on data derived from its own outputs. While watermarks have been extensively studied in diffusion models (DMs) [9, 10, 38, 48], no watermarking methods have been proposed for image autoregressive models such as Infinity, so far. Additionally, existing DM watermarks have been shown to lack radioactiveness [7], motivating the need for a new approach.

To close this gap and provide a robust way of detecting images generated by state-of-the-art autoregressive models operating on bits like Infinity [11] and Instella [37]we propose the BitMark watermark that aligns with bitwise image generation process and operates directly on a *bit-level*. We show that this bit-level design is crucial in image autoregressive models, as watermarks at the token level—a common practice in autoregressive language models [16, 46]—are suboptimal. This is because unlike language models, which operate on discrete tokens and yield stable token representations, image models must decode to and re-encode from a continuous image space, introducing discrepancies between the original and recovered tokens. These discrepancies can decrease the watermark signal. Yet, we find that the discrepancies are substantially smaller at the bit level (Figure 1), motivating our bitwise approach to obtain a robust watermark. To embed the watermark, BitMark first divides all possible *n-bit sequences* into *green* and *red* lists, where the green list is supposed to contain sequences indicating the presence of the watermark. The embedding process of the watermark aims at completing one of the green-list bit sequences for every next bit by slightly increasing each logit of the preferred bit that would complete the sequence. This eventually causes the generated image to contain more sequences from the green list than natural images, which can be used for detection. Concretely, the detection of the watermark is performed on re-encoded images and based on the count of bit sequences from the green list.

Our watermark exhibits many desirable properties. Most importantly, the generated images are of similar quality as images generated without the embedded watermark. Moreover, the watermark embedding has a negligible impact on the speed of the image generation. Also, the corresponding watermark detection method is fast and reliable. As in natural images all bit sequences are equally likely to occur while our watermarked images have the bias towards sequences from the green list, natural images are extremely unlikely to be detected as watermarked. Furthermore, working on the *bit-level* makes BitMark robust against a wide range of attacks, such as adding Gaussian noise or blur to the image, applying color jitter, random crop, or JPEG compression. Our watermark is also highly resilient against dedicated watermark-removal techniques, for example, the watermark in the sand attack [42] or CtrlRegen [20]. Finally, we show that BitMark is radioactive. This means that when we use the watermarked images to train or fine-tune other image generative models, our watermark is still present in the outputs from these other models. We demonstrate all these properties through our extensive experimental evaluation.

In summary we make the following contributions:

- We propose BitMark—the first watermarking scheme for bitwise image generative models. BitMark leverages the bitwise image generation process of these models to embed the watermark on the level of bits, which preserves the high quality of the models' outputs and its inference speed. We show the success of our method on Infinity and Instella.

- We thoroughly evaluate BitMark against a wide variety of standard and dedicated watermark removal attacks and show its robustness to all of them. Specifically, BitMark is also resilient to the watermark in the sand attack [42], which is targeted at removing the watermarks from the generated content. Finally, BitMark is also robust against a Bit-Flipper attack carefully crafted against it.

- We also assess BitMark's *radioactivity* and show that when other image-generative models are trained or fine-tuned on images watermarked with our method, these new models' outputs will also contain our watermark. This makes BitMark highly suitable to identify generated content and contributes to preventing model collapse due to iterative training a model on its generated images.

## 2 Background

We first introduce image autoregressive models, with a focus on the state-of-the-art text-to-image models that operate on bits, namely Infinity and Instellaoperating in an autoregressive manner. We then provide an overview on techniques used to robustly watermark image generative models.

**Visual AutoRegressive Modeling (VAR)** [35]. VAR leverages discrete quantization to predict tokens as residual maps, scaling from an initial low scale (coarse-grain) to a final scale (fine-grain) resolution. VAR further leverages the transformer architecture [36] across all tokens to significantly improve generation quality. VAR was extended from class-to-image to the **Infinity** model [11], which is a text-to-image autoregressive model for high-resolution image synthesis, using next-scale prediction similar to VAR [35]. Instead of leveraging index-wise tokens for quantization like VAR [35], Infinity uses bitwise token prediction with an implicit codebook and applies binary spherical quantization [47] to select an entry from the codebook. This results in more efficient computation and further enables the usage of large codebook sizes, *e.g.,* $2^{64}$. Random bit flipping further empowers the model to self-correct on each scale, leveling out errors stemming from earlier tokens. The visual autoregressive paradigm instantiated by Infinity achieves the state-of-the-art (SOTA) in image generation. **Instella** [37] is a token-lightweight next-patch prediction autoregressive model. It efficiently leverages bits to minimize the number of tokens required for generating high-resolution images, reducing it to as few as 128 tokens. This bitwise paradigm allows Instella to achieve competitive results in image generation with significantly lower computational resources.

**Watermarking Images.** Watermarking entails embedding a message, pattern, or information into an image, that can be detected. The embedded information should be hard to remove and can be also used to verify the provenance of the image. In general, the watermarking methods can be either *static* or *learning-based*. In static watermark embedding, a transform is directly applied to an image [22]. Learning-based methods are generally based on three key components: a watermark ($w$), an encoder ($E$) and a decoder ($D$). The encoder is trained to embed the watermark into the image, while not impacting its quality and the decoder is trained to reconstruct the watermark given a watermarked image to perform detection. An important aspect of watermarks is the trade-off between their detectability and their impact on the quality of the data, which was thoroughly analyzed in [39].

**Watermarking Diffusion Models.** Learning-based watermarks represent the SOTA for Diffusion Models [9, 10, 38, 48]. They target different points of the image generation process, such as the initial data used to train the model [48], noise prediction [10, 38], or the decoding process [9]. The Recipe method [48] embeds a binary signature into training data using a pretrained encoder-decoder and trains an explicit decoder to extract this watermark from outputs. In contrast, Tree-Ring [38] embeds a structured pattern in the initial noise vector, designed in the Fourier space, achieving robustness to transformations by enabling detection through inversion of the diffusion process. On the other hand, the PRC Watermarking [10] generates the initial noise vector from a pseudorandom error-correcting code, allowing deterministic reconstruction of the noise for watermark retrieval. Finally, Stable Signature [9] fine-tunes the latent decoder of latent diffusion models (LDMs) to insert a recoverable watermark using a pretrained extractor, modifying the generation process at the decoding stage. However, the watermarking techniques for diffusion models are not radioactive [7], hence, do not prevent the model collapse.

**Watermarking Autoregressive Models.** As of now, all the efforts on watermarking autoregressive models have been directed towards creating watermarks for Large Language Models (LLMs) [16]. The KGW watermark [16] uses the concept of a green and red token lists, where generation is softly nudged to predict tokens from the green list rather than from the red list. The lists are generated dynamically, based on the hash value of the previously predicted token(s). For detection, the LLM used to generate the text is not needed. Instead, a suspect text is assessed based on how often tokens from the green list appear, compared to tokens from the red list. At the end of detection a statistical test is performed. KGW was extended to the Unigram-Watermark [46], which simplifies the design by introducing a fixed global green and red lists. The simplified approached allowed for guaranteed generation quality, provable correctness in watermark detection, and robustness against both text editing and paraphrasing. In this work, we build on these watermarking schemes originally developed for large language models to address the lack of watermarking techniques for bitwise generative models.

**Private Watermarking.** In our watermark, we follow the *Private Watermarking* from KGW [16]. For this end, the generative model is kept behind a secure API with a secret key $\mathcal{K}$. This secret key $\mathcal{K}$ is used as input to a pseudorandom function (PRF) $F$, with the $h$ previously generated tokens, to compute the red-green list split for the next predicted token. Here the context window $h$ is a hyperparameter, which trades robustness with privacy. For small context windows $h$ (*e.g.,* 1 or 2) the watermark is robust against removal attacks but an attacker could tabulate the frequency of token pairs and brute-force knowledge about the green list, reducing the privacy of the method. For larger values of $h$ (*e.g.,* 5), changing a single token has an impact on the next $h$ downstream tokens, potentially flipping them to the red set. The privacy of this approach can also be boosted by using $k$ different private keys and choosing one randomly during generation.

**Watermark Robustness.** To reliably perform image provenance, a watermark must be robust to removal attacks. There is a wide range of these kinds of attacks that have been proposed in the literature [2, 20, 27]. For example, the scrubbing attacks aim at removing the watermarks from the outputs of a generative model, which can be done by producing a valid *paraphrase* that is detected as non-watermarked text [15] or image [3]. Zhang et al. [42] claim that under the assumption that both a quality oracle (that can evaluate whether a candidate output is a high-quality response to a prompt) and a perturbation oracle (which can modify an output with a nontrivial probability of maintaining quality) exist, it is feasible to remove existing watermarks in generative models without quality degradation of the watermarked content. We evaluate our BitMark against a wide spectrum of attacks and show that our method is robust against them.

## 3 BitMark Method

The autoregressive image generative models that we consider, namely Infinity [11] and Instella [37], predict bits instead of tokens, distinguishing them from other image autoregressive models such as VAR [35] or RAR [41]. We leverage this prediction scheme to design a dedicated watermark. Our *BitMark* watermark is embedded during the generation process, subtly shifting the bitwise representation towards a higher frequency of detectable patterns to resist manipulation.

**Problem Formulation and Desiderata.** Our goal is to embed a watermark during the image generation process and subsequently detect it by analyzing the generated image. BitMark targets the prevention of model collapse. In this setup, only the model owner is concerned of detecting the watermark and wants to ensure that their own data are not used to train their next models. Thus, we aim at four essential properties for our watermarking scheme to prevent model collapse: (1) ***Negligible Impact on the Generation.*** To maintain the utility of the generative model, the watermark should not degrade the quality of the generated images, while also maintaining a fast inference speed. (2) ***Reliable Verification.*** The model owner should be able to detect the watermarked images and verify the confidence of the extracted watermark. The model owner encodes the image to their own space and it is only relevant to the model owner if the watermark is present there or not. The encoder part of the model is also kept private so only the model owner can verify the watermark. The natural images should not be detected as containing the watermark. At the scale of the large models trained on trillions of samples, a small number of false positives is acceptable as long as it does not exceed a certain threshold (*e.g.,* 1%), which can also prevent model poisoning [45]. (3) ***Robustness.*** The watermark should be robust not only against conventional attacks, such as Gaussian noise or JPEG compression, but also against strong dedicated watermark removal attacks such as CtrlRegen [20] or watermarks in the sand [42]. (4) ***Radioactivity.*** If other generative models are trained on watermarked images, images generated by these new models should also contain our watermarks [7, 28, 31].

**Threat Model.** We assume two parties: a *model owner* providing an image generation API where our BitMark is deployed, and an *attacker* who aims to fool the model owner to train on their own generated images and to prevent the use of large-scale natural image datasets for model training. We assume that the **model owner** follows the *Private Watermarking* from [16] and keeps the generative model behind a secure API. The model owner wants to watermark their own images in order not to train on them. The watermark should also be radioactive so that if any other party fine-tunes their own models on the watermarked data, the fine-tuned model's outputs are watermarked, as well. Further, we assume that the watermarking technique is private and verifiable by the model owner. On the other hand, the **attacker** might attempt to remove the embedded watermark. For instance, competitors might still want to remove the watermark to harm the model owner. We assume that the attacker has black-box access to the model, *i.e.,* they can provide inputs and observe the corresponding outputs. The attacker is allowed to make arbitrary modifications to the generated image, including

but not limited to: applying image transformations (*e.g.,* noising, blurring, cropping, etc.) or specific watermark removal techniques [20, 42]. We define adversarial behavior as any attempt to remove the watermark in the generated image while preserving high perceptual quality and semantic content. Attacks that destroy the usefulness or visual coherence of the image (*e.g.,* replacing it with noise) are not considered meaningful threats.

## 3.1 Motivation for the Bitwise Watermark

We identify an **inherent challenge** in watermarking the images generated by autoregressive models stemming from imperfections in image encoding and decoding. Specifically, the model first generates a sequence of tokens, which are decoded into an image. However, when the exact same image is re-encoded and tokenized, it produces a different set of tokens. This also holds partly true for bits that underlie the tokens. The observed behavior does not occur in the autoregressive LLMs, where the tokens produced by decoding and then re-encoding a generated text match *exactly*. We quantify the bit and token discrepancies in Figure 1 for Infinity. In both cases, bits and tokens are generated by the model and then reconstructed by its auto-encoder. Here, the highest bit overlap occurs at the initial and final resolutions (scales). This is likely because the early stages capture coarse structural features, while the final resolution refines the image, leading to fewer changes in the bit sequence. We observe a consistent average token overlap of approximately 2.385% and bit overlap of 77.432%. Given the significantly large codebook size of $2^{32}$ in Infinity, we observe that the bitwise information loss is much smaller than the

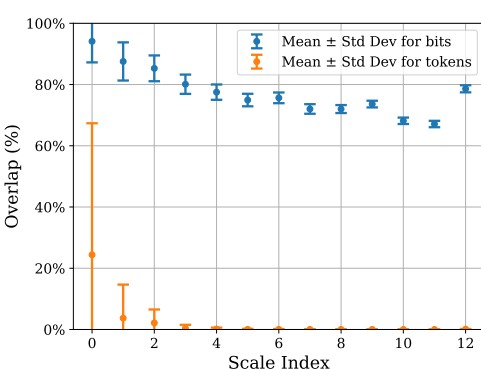

Figure 1: **Bit and token overlaps in Infinity.**

token-wise loss. This motivates our work to design a watermark that biases the generated sequences of tokens on the *bit-level* instead of on the *token-level*.

## 3.2 Embedding our BitMark Watermark

Next, we detail how BitMark is embedded. The embedding protocol is directly applied on the *bit-level* in the autoregressive generation process of the model. We present the full procedure for multi-scale models like Infinity in Algorithm 1 and for single-scale models like Instella IAR in Algorithm 3. First, we separate the list of all bit sequences of length $n$: $S = \{0,1\}^n$ into a red list $R$ and a green list $G$. Intuitively, we softly nudge the model to generate bit sequences of length $n$ that are part of the green list $G$. For each next bit prediction $b_j$, if the bit can complete one of the bit sequences from $G$, we add a small bias $\delta$ to the logit $l_j^{(b_j)}$ of the bit $b_j$:

$$p_j = \begin{cases} \frac{\exp(l_j^{(b_j)}+\delta)}{\exp(l_j^{(\neg b_j)})+\exp(l_j^{(b_j)}+\delta)}, & pre + b_j \in G \\ \frac{\exp(l_j^{(\neg b_j)})}{\exp(l_j^{(\neg b_j)})+\exp(l_j^{(b_j)}+\delta)}, & pre + \neg b_j \in R, \end{cases} \tag{1}$$

where $pre = (b_{j-(n-1)}, \ldots, b_{j-1})$ is the prefix bit sequence of length $(n-1)$, *i.e.,* the sequence of bits preceding $b_j$, $+$ is the bit concatenation operation, and $l_j$ is the vector of logits for the bit $b_j$. We apply this procedure to all bits that are generated by the model. Thus, as many bit sequences as possible are biased towards $G$ and the final green fraction, *i.e.,* the number of sequences part of the green set, of the generated bits is significantly above the threshold of 50%, the standard fraction for clean images (see Section F.6). By biasing the logit for the preferred bit, BitMark has negligible impact on image quality, as mostly high entropy bits are flipped (see Table 1). Since high entropy bits have sufficiently small difference in the probability of assigning them to value 0 or 1, the relatively small added bias influences the final value. Additionally, biasing only the logits makes the watermarking process difficult to revert, as it is difficult to know for an attacker which bits are of high entropy and can be flipped to the red list $R$ without degrading the final image quality (see Figure 9).

## 3.3 Watermark Detection

Watermark detection is performed directly on the bits of encoded images. All that is required is knowledge about the green and red lists, as well as access to the tokenizer. Intuitively, we count how often each sequence of bits in the encoded image follows the green list and compare how the frequency differs from non-watermarked images.

**Detecting the Watermark.** The detection protocol follows the standard encoding of the model and is detailed for multi-scale models in Algorithm 2 and for single-scale models in Algorithm 4. First we obtain the bit representations for each resolution. We then use our knowledge of $G$ and $R$ to test how often the bit sequences from the red and green lists appear and count them. This results in the knowledge of how many green sequences appear in the encoded image, which is random for every non-watermarked image, *i.e.,* $50\%$ (see Section F.6).

**Robust Statistical Testing.** Given the final number of green and red sequences detected in the encoded image, we perform a statistical test based on the null hypothesis $\mathcal{H}_0$ : *The sequence of bits is generated with no knowledge about the green and red lists.* Intuitively, this assumes that a given image is not watermarked. A watermarked image consists of $\gamma T$ green and $(1 - \gamma)T$ red bit sequences, where $\gamma$ determines the size of the green list and $T$ is the total number of bits in the encoded image that our watermark can influence. Per a given token $t$ with $m$ bits, BitMark influences $k = m - (n - 1)$ bits, since we start watermarking from the first $n$ bit pattern. Thus, overall we have $T = \sum_{i=1}^{K} r_i \cdot k$, where $r_i$ is number of tokens per resolution $i$. A natural image is expected to violate the red list rule with the half of its bits, while a watermarked image violates this rule less frequently. If the null hypothesis is true, then the number of green list bit sequences denoted C has the expected value of $\gamma T$ and variance $\gamma(1 - \gamma)T$. To ensure statistical validity, we follow [16] and use a z-test:

$$z = (C - \gamma T)/\sqrt{T\gamma(1 - \gamma)} \tag{2}$$

We reject the null hypothesis and detect the watermark if $z$ is above a chosen threshold.

---

**Algorithm 1** Watermark Embedding

**Inputs:** autoregressive model $f_\theta$ with parameters $\theta$, green list $G$, red list $R$, image decoder $\mathcal{D}$.
**Hyperparameters:** steps $K$ (number of resolutions), resolutions $(h_i, w_i)_{i=1}^{K}$, the number of tokens for resolution $i$ is $r_i$, constant $\delta$ added to the logits of the *green* list bits, $n$ - the length of the bit sequences in the lists ($G$ and $R$), $m$ the number of bits per token.
**for** $i = 1, \ldots, K$ **do**
    $(l_1, \ldots, l_{r_i \cdot m}) = f_\theta(e, h_i, w_i)$
    **for** $t = 1, \ldots, r_i$ **do**
        **for** $j = n, \ldots, m$ **do**
            $c = (t - 1) \cdot m + j$ //counter
            $p_c = \text{Bias}(G, l_c, \delta)$
            $s_c = \text{Softmax}(p_c)$
            $b_c = \text{Sample}(s_c)$
    $u_i = (b_1, \ldots, b_{r_i \cdot m})$
    $z_i = \text{Lookup}(u_i)$
    $z_i = \text{Interpolate}(z_i, h_K, w_K)$
    $e = e + \phi_i(z_i)$
$im = \mathcal{D}(e)$
**Return:** watermarked image $im$

---

**Algorithm 2** Watermark Detection

**Inputs:** raw image $im$, green list $G$, red list $R$, image encoder $\mathcal{E}$, quantizer $\mathcal{Q}$
**Hyperparameters:** steps $K$ (number of resolutions), resolutions $(h_i, w_i)_{i=1}^{K}$, the number of tokens for resolution $i$ is $r_i$, $n$ - the length of the bit vector.
$e = \mathcal{E}(im)$
$C = 0$
**for** $i = 1, \ldots, K$ **do**
    $u_i = \mathcal{Q}(\text{Interpolate}(e, h_i, w_i))$
    $u_i = (b_1, \ldots, b_{r_i \cdot m})$
    $C = \text{Count}((b_1, \ldots, b_{r_i \cdot m}), G)$
    $z_i = \text{Lookup}(u_i)$
    $z_i = \text{Interpolate}(z_i, h_K, w_K)$
    $e = e - \phi_i(z_i)$
**Return:** StatisticalTest($\mathcal{H}_0, (C)$)

---

## 4 Empirical Evaluation

**Experimental Setup.** In the following, we employ the Infinity-2B checkpoint [11] as the default model to evaluate BitMark across different watermarking strengths and against a wide variety of attacks. Additionally we show the generalizability of BitMark to other models, namely the Infinity-8B checkpoint as well as the Instella-T2I [37] IAR model. We further show generalizability of our method to bitwise diffusion models in Section G. The Infinity models natively generate $1024 \times 1024$

pixel RGB-images, we choose the visual-tokenizer with a vocabulary size of $2^{32}$ and $2^{56}$ per token for the Infinity-2B and the Infinity-8B checkpoint, respectively. We follow Han et al. [11] in the choice of hyper-parameters and use a classifier-free-guidance of 4, with a generation over 13 scales, following the scale schedule for an aspect ratio of 1:1. Hence, we utilize the best performing model, tokenizer and hyperparameters reported by Han et al. [11]. We ablate the choice of the green and red list split for our experiments in Section D. For Instella IAR we follow Wang et al. [37], leveraging a vocabulary of size $2^{32}$ and a cfg of 7.5 at a $1024 \times 1024$ resolution. We evaluate our experiments based on images and prompts of the MS-COCO 2014 [18] validation dataset. Unless stated otherwise, we use a subset of 10,000 images.

**Baselines.** To the best our knowledge no prior work developed an in generation-time watermarking scheme for autoregressive image models. Therefore we compare the robustness of BitMark against three postprocessing watermarks in Table 2. Namely we analyze RivaGAN [43] (with message length = 32), StegaStamp [34] (with message length = 100) and TrustMark [5] (with message length = 100). We applied the postprocessing watermarks on 1,000 images generated by Infinity-2B.

## 4.1 BitMark Introduces Detectable Watermarks while Maintaining Generation Performance

We demonstrate that our BitMark preserves the utility of the model with negligible impact on the image quality and the inference speed.

**High Image Quality.** Table 1 presents a quantitative evaluation of image generation quality for different watermarking strengths dependent on the logit bias parameter $\delta$. We report the FID, KID, measuring the similarity of generated images and natural images in feature space, and the CLIP Score, which measures the alignment between prompt and image, for the varying watermark strengths $\delta$.

The results show that, for $\delta \leq 3$, BitMark has a negligible impact on image quality (as measured by FID and KID) and minimally affects the prompt-following capabilities of the underlying model, as indicated by the CLIP Score. We also present a qualitative analysis of image quality in Section F.3, which shows images generated for different $\delta$. While there is an inherent trade-off in terms of watermark strength an image quality, we choose a small $\delta = 2$ for the Infinity-2B checkpoint to obtain minimal im-

Table 1: **Image quality mean (std) vs $\delta$.**

| $\delta$ | FID$\downarrow$ | KID ($\times 10^{-2}$)$\downarrow$ | CLIP Score$\uparrow$ |
|---|---|---|---|
| 0 | 33.36 | 1.42 (0.12) | **31.16 (0.28)** |
| 1 | 32.61 | 1.38 (0.13) | **31.16 (0.28)** |
| 2 | 31.05 | 1.26 (0.12) | 31.15 (0.28) |
| 3 | **29.61** | **1.03 (0.08)** | 31.03 (0.27) |
| 4 | 42.98 | 1.78 (0.08) | 29.78 (0.32) |
| 5 | 127.44 | 11.16 (0.34) | 26.13 (0.40) |

pact on the generation while having a significantly strong watermark. We further show that $\delta = 1.5$ does not negatively impact image quality for Infintiy-8B and Instella in Table 15.

**Fast Watermark Generation and Verification.** For a watermark to be reasonably applicable in a real world setting, the watermark should also have negligible impact on image generation speed [9, 38, 48]. We thus analyze the practicality of applying BitMark in a standard setting and its impact on generation speed of the Infinity model. Next to the generation performance, the speed of watermark detection has a strong impact on the possible applications, especially in a setting where we want to quickly detect generated data. The quantitative analysis of BitMark in Section F.4 shows that generating a watermarked image requires at most one-tenth the time longer than the standard image generation. Verification, *i.e.,* encoding an image into its bit representation and computing the $z$-score, takes less than 0.5 seconds per image, usually around $5\times$ faster than the image generation.

## 4.2 BitMark is Robust Against a Wide Range of Attacks

Our results demonstrate that BitMark is robust against a wide range of attacks, including conventional attacks, reconstruction attacks, watermarks in the sand attack, and our own adaptive *bit-flipping* attack, with the aim of exploiting knowledge of our watermark method to remove BitMark.

**Robustness against Conventional Attacks.** We perform a wide range of conventional attacks, namely: 1) **Noise:** Adds Gaussian noise with a std of 0.1 to the image, 2) **Blur** Noise attack + application of a Gaussian blur with a kernel size of 7, 3) **Color**: Color jitter, a random hue of 0.3, saturation scaling of 3.0 and contrast of 3.0, 4) **Crop:** Random crop to 70%, 5) **Rotation:** Random rotation between 0° and 180°, 6) **JPEG:** 25% JPEG compression, 7) **Horizontal Flipping**, 8) **Vertical Flipping**.

Table 2: **BitMark is robust against watermark removal attacks.** We report the TPR@1%FPR (%) for the different conventional and reconstruction attacks.

| Watermark | Conventional Attacks | | | | | | | | | Reconstruction Attacks | |
| | None | Noise | Blur | Color | Rotate | Crop | JPEG | Vertical | Horizontal | SD2.1-VAE | CtrlRegen+ |
|---|---|---|---|---|---|---|---|---|---|---|---|
| RivaGAN [43] | 99.7 | 98.3 | 99.7 | 99.4 | 96.7 | 99.4 | 99.7 | 0.0 | 0.0 | 98.5 | 1.6 |
| StegaStamp [34] | 100.0 | 100.0 | 100.0 | 98.7 | 32.1 | 1.0 | 100.0 | 1.0 | 33.8 | 100.0 | 44.2 |
| TrustMark [5] | 99.9 | 99.5 | 99.9 | 2.2 | 2.7 | 1.6 | 99.9 | 0.7 | 99.8 | 99.7 | 1.1 |
| Infinity-2B ($\delta = 2$) | 100.0 | 99.6 | 99.9 | 99.8 | 20.1 | 98.8 | 100.0 | 78.8 | 100.0 | 100.0 | 91.6 |
| Infinity-8B ($\delta = 1.5$) | 100.0 | 99.7 | 100.0 | 75.5 | 57.6 | 99.4 | 99.9 | 93.8 | 99.7 | 100.0 | 30.4 |
| Instella IAR ($\delta = 1.5$) | 100.0 | 96.0 | 100.0 | 75.3 | 2.7 | 7.5 | 100.0 | 9.3 | 12.1 | 100.0 | 93.6 |

Table 2 shows that our approach achieves a high TPR@1%FPR for most conventional attacks compared to the baselines. Each baseline is weak against at least one conventional attacks. RivaGAN is not robustness against flipping (0%TPR@1%FPR), StegaStamp is not robustness against cropping (1%TPR@1%FPR) and TrustMark is vulnerable to color jitter (2.2%TPR@1%FPR), rotation (2.7%TPR@1%FPR) and cropping (1.6%TPR@1%FPR). Although Bit-

Table 3: **Rotation attack.**

| Degrees | TPR@1%FPR | $z$ (std) |
|---|---|---|
| -20, 20 | 99.9 | 14.87 (9.77) |
| -30, 30 | 87.4 | 11.78 (9.55) |
| -40, 40 | 70.0 | 9.66 (9.38) |
| -50, 50 | 56.8 | 8.21 (8.99) |

Mark for Infinity-2B has slightly lower performance towards arbitrary rotation angles, the watermark is still highly detectable and it obtains high TPR@1%FPR within a given rotation threshold. We report TPR@1%FPR and mean (std) of $z$ for different degrees of rotations for 1,000 images in Section F.2. Notably, it achieves as high as 99.9% TPR@1%FPR when the rotation range is between -20° and 20° as shown in Table 3. Given the strength of off-the-shelf tools for rotation estimation [21], it is highly achievable to rotate the image back into a tolerable range of rotations during the pre-processing stage of the detection, resulting in a TPR@1%FPR of 98.3% for randomly rotated images as shown in Table 13. Table 12 provides an ablation of the robustness of BitMark given different values of $\delta$, showing that BitMark for Infinity-2B is highly robust against most attacks even with $\delta = 1$. Infinity-8B is more robust against rotation compared to the smaller 2B checkpoint. The effectiveness of BitMark on Instella IAR is by design very limited, given its total number of 128 tokens, corresponding to 4,096 bits, per image.

**Robustness against Reconstruction Attacks.** We also evaluate our proposed method against two state-of-the-art reconstruction attacks (shown as the last two columns in Table 2): 1) **SD2.1-VAE:** Encoding and decoding with the Stable Diffusion 2.1 autoencoder [26], 2) **CtrlRegen+:** Application of controllable regeneration from noise in the latent space following [20]. Regarding reconstruction attacks, our results indicate that BitMark for Infinity-2B is robust against the CtrlRegen+ reconstruction attack, achieving 91.6%TPR@1%FPR. Also Instella IAR achieves strong robustness of 93.6%TPR@1%FPR. In contrast, RivaGAN and TrustMark demonstrate a lack of robustness with 1.6%TPR@1%FPR and 1.1%TPR@1%FPR, respectively. StegaStamp exhibits partial robustness of 44.2%TPR@1%FPR. The Infinity-8B shows only partial robustness, which is further discussed in Section A.

Table 4: **BitMark is robust against the watermarks in the sand attack** [42]. We present the detection results of our watermarking scheme ($\delta = 2$) before and after the attack. We use 200 iterations, and generate 100 candidates during each iteration.

| Setting | Accuracy (%) | AUC (%) | TPR@1%FPR (%) | Green Fraction (%) | $z$ | $z$ Threshold @1%FPR |
|---|---|---|---|---|---|---|
| Original Image | - | - | - | 49.7 | 0.2 | - |
| Watermarked Image (before attack) | 100.0 | 100.0 | 100.0 | 57.6 | 87.8 | 59.6 |
| Watermarked Image (after attack) | 98.5 | 99.8 | 89.0 | 50.9 | 10.8 | 6.8 |

**Robustness against Watermarks in the Sand Attack.** We also evaluate the robustness of our watermarking scheme against the attack proposed in [42], namely, the watermarks in the sand attack. The attack leverages the CLIP Score as a quality oracle, the stable-diffusion-2-base [23] as a

perturbation oracle, and iteratively modifies the watermarked image to remove the watermark. During each iteration, the attack produces potential candidates by inpainting randomly masked regions in the image. The attack successfully replaces most visual details in the original watermarked image while selecting the candidate with the best CLIP Score. As a result, many watermarks for diffusion models (*e.g.,* [9]) can be removed through this attack. However, results in Table 4 show that our proposed method achieves a significant detection rate with TPR@1%FPR of 89.0%. Although a large percentage of the green-list patterns are removed, the average $z$-score in the attacked images is still as high as 10.8, which remains significant when compared with a mean $z$-score of 0.2 for non-watermarked images. Moreover, the attack causes a quality loss of the image when compared to the original image generated by Infinity. With 200 iterations and 100 candidates during each iteration, the method further requires a high computational cost, where more than 12 minutes are needed to attack a single image in the environment specified in Section E.2. We note that the results do not conclude the failure of the theoretical framework in watermarks in the sand, but rather that there are currently no successful instantiations of the perturbation oracles and quality oracle for Infinity that can remove our proposed watermark. A more detailed analysis can be found in Section H.2.

**Robustness Against our Novel Bit-Flipper Attack.** We further demonstrate that, even giving the attacker much stronger knowledge than our threat model formulated in Section 3, BitMark cannot be erased even with severe image degradation. Concretely, we assume the attacker is given the knowledge of the auto-encoder of the generative model, the watermark embedding and detection algorithms, and even our red and green lists. We propose a novel Bit-Flipper attack tailored for this broader attack surface, aiming at removing our watermarks. During this attack, the attacker first computes the total number of bit sequences $|s|$ and calculate the green fraction $C/|s|$. Then, the attacker randomly flips bits to convert green bit sequences to red

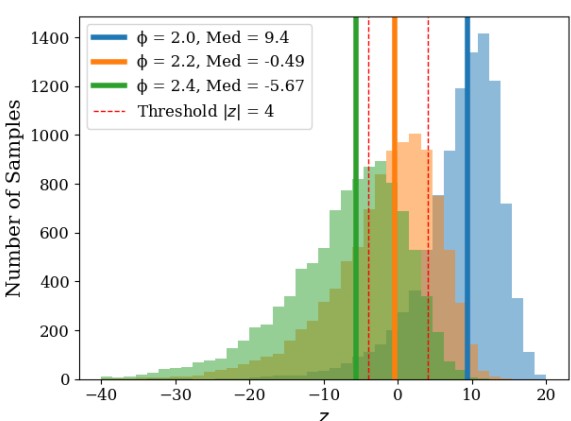

Figure 2: **Bit-Flipper attack** for $\delta = 2$

ones. The goal of the bit flipping is to reduce the green fraction to the level of non-watermarked images, which is 50%, as shown in Section F.6. Specifically, the attacker computes the probability of flipping a bit via a flip factor $\phi$, such that $p = (C/|s|) - 0.5) * \phi$, whereas $\phi$ determines the bit sequence removal severity. A detailed algorithm of our attack is described in Algorithm 5. As shown in Section H.1, $\phi = 2.2$ has the most impact on detectability. However, randomly flipping bits comes with very high degradation in image quality, as shown in Figure 10. Due to the symmetric detection threshold $|z|$, if we destroy too many bit sequences in $G$, this further leads to the detectability of the watermark (see Section H.1).

## 4.3   Our BitMark Watermark Exhibits High Radioactivity

We demonstrate that our BitMark exhibits high radioactivity, *i.e.,* other generative models fine-tuned on our watermarked images also produce our bit patterns that are reliably detectable.

**Formalizing Radioactivity.** We denote by model $M_1$ our original model that generates watermarked outputs with a watermarking method, *e.g.,* our BitMark. These watermarked outputs might be used as a training set for a second model $M_2$. In the context of language models it has been shown, that there are watermarks that persist this process and remain detectable in the outputs of $M_2$, a property referred to as *radioactivity* [31]. This property enables the detection of generated content, even when it has been used to train subsequent models. However, recent findings indicate that this property does not hold for watermarks in image generative models, such as diffusion models [7].

**Training Setup.** Given the extensive compute costs of training high-performant image-autoregressive models from scratch, we simulate full training by full fine-tuning the models. Therefore, we first generate 1,000 images with prompts of the MS-COCO2014 [18] validation dataset with our BitMark embedded under a watermarking strength of $\delta = 2$. The watermarked outputs of $M_1$ are then used to

Table 5: **BitMark exhibits strong radioactivity.** We report the TPR@1%FPR (%).

| Type of $M_1$ | Type of $M_2$ | Output of $M_1$ | Output of $M_2$ |
|---|---|---|---|
| Infinity-2B | Infinity-2B | 100.0 | 100.0 |
| Infinity-2B | Stable Diffusion 2.1 | 100.0 | 98.9 |

train (*i.e.,* in our approximative setup full fine-tune) the $M_2$ model. We analyze two different settings, with the same $M_1$ of Infinity-2B, namely: (1) the same modality setting, transfer to Infinity-2B and (2) the cross-modality setting, transfer to the Latent Diffusion Model (LDM) Stable Diffusion 2.1 [26]. To test the radioactivity, we compute the TPR@1%FPR of output images of $M_2$ and report them in Table 5.

**BitMark is Highly Radioactive.** Table 5 highlights the strong radioactivity of our BitMark. When fine-tuning a model $M_2$ of the same architecture, the watermark is still detectable with a 99.7%TPR@1%FPR. Even in the cross-modality setting, BitMark is still detectable with a 98.9%TPR@1%FPR. Dubiński et al. [7] attribute the lack of radioactivity of watermarks in LDMs to the encoding from the pixel space into the latent space and the *noising-denoising* process while training $M_2$.

However, our BitMark remains detectable despite the encoding and the noising-denoising process. The robustness of our watermark arises from slightly changing all parts of a given generated image, including low-level and high-level features, which are then also partly adapted by $M_2$. The observed results are in line with our robustness against CtrlRegen+ (see Table 2), an attack which simulates noising and denoising in the latent space. BitMark's radioactivity is a very useful property that allows model owners to track provenance of their data and data generated based on it, even when this new data does not originate *directly* from their original model.

We additionally ablate the ratio of watermarked to clean data needed for our BitMark to become radioactive to the Inifinity-2B model in Table 6. Even when model $M_2$ is trained on only 10% of our watermarked data, it generates images with a clear bias towards our watermark distribution, resulting in a high TPR of 22.7% at 1% FPR. We note that even a slight bias is significant at the distribution level. To validate this claim, we apply a one-tailed Mann-Whitney U test to determine if the z-score distribution of $M_2$-generated data is significantly larger than on $M_1$ clean generated data. The test is significant with a p-value of 5.6e-62 when $M_2$ is trained on only 5% watermarked data, showing that our BitMark provides a traceable signal across training cycles in settings where only small fractions of the training data are watermarked.

Table 6: **Detection performance and statistical significance across fractions of watermarked data.** TPR@1%FPR (top row) and Mann–Whitney U-test p-values (bottom row) for different fractions of watermarked data used for fine-tuning.

| Metric \ p% of watermarked data | Clean model | 0 | 1 | 5 | 10 | 25 | 50 | 100 |
|---|---|---|---|---|---|---|---|---|
| TPR@1%FPR | 1.0% | 0.3% | 1.0% | 2.9% | 22.7% | 98.3% | 100.0% | 100.0% |
| Mann–Whitney U Test (p-value) | 0.999 | 0.999 | 0.999 | 5.6e-62 | 6.8e-209 | $\approx 0$ | $\approx 0$ | $\approx 0$ |

## 5   Conclusions

We present BitMark, the first watermarking framework for bitwise image autoregressive models. We show that our watermark can be efficiently embedded and detected, imposes negligible overhead on image quality and generation speed, while exhibiting a strong robustness against a wide range of watermark removal attacks, including the strong watermarks in the sand and our own adaptive Bit-Flipper attack. Moreover, BitMark exhibits strong radioactivity, *i.e.,* downstream models fine-tuned on our watermarked outputs consistently inherit and reproduce our bit patterns, which makes our watermark reliably detectable requiring only 5% watermarked data to achieve a significant distributional shift. This is important for model owners to identify images derived from their own generated images, and exclude them from training of future model versions. Overall, BitMark not only sets a new standard for watermarking in autoregressive image generation, but also provides a critical defense against model collapse, contributing to further advancements in image generative models.

## Acknowledgments and Disclosure of Funding

This work was supported by the German Research Foundation (DFG) within the framework of the Weave Programme under the project titled "Protecting Creativity: On the Way to Safe Generative Models" with number 545047250. Responsibility for the content of this publication lies with the authors. We would like to also acknowledge our sponsors, who support our research with financial and in-kind contributions, especially the OpenAI Cybersecurity Grant.

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

## A   Limitations and Broader Impact

**Limitations.**   While BitMark for Infinity-2B is robust against moderate rotations (99.9% TPR@1%FPR within rotation angles -20° to 20°), the detection accuracy degrades with larger rotation angles, dropping to 56.8% TPR@1%FPR for ±50° rotations. Although open-source methods have a strong capability for rotation correction, this can be a limitation for real-world scenarios. This rotation sensitivity, however, is an inherent issue in Infinity's data augmentation pipeline, which makes the autoencoder less robust to large rotations. Moreover, the soft biasing parameter $\delta$ requires tuning, as values exceeding $\delta = 3$ lead to noticeable quality degradation (FID increases from 29.61 to 127.44 when $\delta$ increases from 3 to 5), constraining the maximum achievable watermark strength. Although we evaluate the detection BitMark with a commonly adopted metric, *i.e.,* true positive rate at very low false positive rates (1%), it could potentially become problematic in extreme-scale applications involving trillions of samples, where even small false positive rates might impact training data quality. It is observed that the Infinity-8B checkpoint shows increased sensitivity to larger values of $\delta$. We hypothesize that the lower $\delta$ chosen may account for its comparatively lower robustness relative to the Infinity-2B checkpoint, as illustrated in Table 2.

**Broader Impact.** Image generative models have the capabilities of generating photorealistic images, which are near imperceptible to human generated content. BitMark provides a method to watermark the content, without harming image generation capabilities of the model and allows for provenance tracking of generated images for models hosted under a private API. It supports the defense against model collapse, when training a model on data generated by itself.

## B   Additional Related Work

**LLM Watermarking.**   Here, we review more related work regarding the LLM watermarking, which we build upon in our method. Beyond the soft biasing approach proposed by KGW [16], SynthID-Text [6] introduces tournament sampling during the model inference to enhance watermark detectability. This sampling strategy modifies the token sampling process to embed watermark signals. However, tournament sampling is only applicable to autoregressive models with large vocabulary sizes, making it incompatible with the binary bit sampling process used in Infinity. There are also works that incorporate watermarks by training or finetuning the models. For example, REMARK-LLM trains a learning-based encoder for watermark insertion and a decoder for watermark extraction, while using a reparameterization module to bridge the gap between the token distribution and the watermark message [44]. Moreover, potential malicious transformations are incorporated into the training phase to enhance the robustness of the watermarking. Although effective, these training-based methods require substantial computational resources and time. Therefore, we focus on inference-time watermarking methods that can be applied to Infinity without any retraining.

Sander et al. [30] investigate the *radioactivity* of LLM watermarking, which is the ability of watermarks to transfer across different model architectures. They design a specific protocol for amplifying the watermark signals after transferring across models. The radioactivity of watermarks also falls into the scope of our work, as our proposed watermark exhibits significant radioactivity across various model architectures. Regarding the undetectability of the watermark, recent studies show that the biases between watermarked and non-watermarked outputs from the LLM are identifiable with carefully designed prompts [19]. As a specialized application of LLM watermarking, Li et al. [17] design a watermarking scheme tailored for quantized LLMs. The proposed watermarks are only detectable on quantized models while remaining invisible for full-precision models.

**Model Collapse.** With the fraction of artificial generated content within the World Wide Web increasing, Shumailov et al. [33] investigate the phenomenon of model collapsing, which is defined as the decrease of generation quality when models are recursively trained on (partly) generated data. This decrease in generation quality is mainly characterized in overestimating probable events and underestimating low-probability events, as well as recursively adding noise (*i.e.,* errors) which is generated during earlier model generations. Hence, this generated data is polluting future datasets, decreasing the potential quality of successor models and therefore highlighting the necessity to distinguish human- from artificial-generated data. BitMark mitigates model collapsing by robustly watermarking generated images to exclude said marked images from future datasets.

## C    Application of BitMark on Single-Scale models

We report the application of BitMark for single scale models in Algorithm 3 and the detection in Algorithm 4.

We report the functions used in Algorithm 1, Algorithm 2, Algorithm 3 and Algorithm 4 here:

- Bias: Addition of the bias $\delta$ to the current logit $l_c$ based on the green list $G$.
- Sample: Use the probabilities in $p_c$ to sample the returned bit, either 0 or 1.
- Lookup: Perform the BSQ lookup [47] to return the quantized states.
- Interpolate: Scaling to target resolution.
- Count: Counts the number of bits which are part of the green list.

---

**Algorithm 3** Watermark Embedding

**Inputs:** autoregressive model $f_\theta$ with parameters $\theta$, green list $G$, red list $R$, image decoder $\mathcal{D}$.
**Hyperparameters:** the number of tokens $r$, constant $\delta$ added to the logits of the *green* list bits, $n$ - the length of the bit sequences in the lists ($G$ and $R$), $m$ the number of bits per token.
**for** $t = 1, \ldots, r$ **do**
   **for** $j = n, \ldots, m$ **do**
      $c = (t - 1) \cdot m + j$ //counter
      $p_c = \text{Bias}(G, l_c, \delta)$
      $s_c = \text{Softmax}(p_c)$
      $b_c = \text{Sample}(s_c)$
$u = (b_1, \ldots, b_{r \cdot m})$
$e = \text{Lookup}(u_i)$
$im = \mathcal{D}(e)$
**Return:** watermarked image $im$

---

**Algorithm 4** Watermark Detection

**Inputs:** raw image $im$, green list $G$, red list $R$, image encoder $\mathcal{E}$, quantizer $\mathcal{Q}$
**Hyperparameters:** the number of tokens $r$, $n$ - the length of the bit vector.
$e = \mathcal{E}(im)$
$C = 0$
$u_i = \mathcal{Q}(e)$
$u_i = (b_1, \ldots, b_{r \cdot m})$
$C = \text{Count}((b_1, \ldots, b_{r \cdot m}), G)$
**Return:** StatisticalTest($\mathcal{H}_0, (C)$)

---

## D    Green and Red List Selection

**Consistency in Longer Sequences.** We analyze the overlap of bit sequences with growing size in Figure 3 which shows that with increased length of the sequence, the consistency of the pattern decreases. This has a direct impact on the choice of $n$, as it shows that larger $n$ lead to less overlap, reducing the impact of BitMark which is already noticeable for $n = 3$ (see Table 7).

**Selection of Red and Green Lists.** The re-encoding loss presented in Section 3.1 also motivates the choice of a small length of the green list $n$, as larger consecutive bit sequences are more likely to be interrupted. Additionally, this auto-encoder has the property of re-encoding images towards an equal frequency of 0's and 1's, so it is not efficiently possible to bias each bit towards either 0 or 1, making the choice of $n = 1$ unpractical (see Table 7). Thus we leverage the second smallest possible choice, $n = 2$, resulting in 6 possible choices to separate $G$ and $R$ by keeping $|G| = |R|$.

As we have the constraint to not bias towards 0 or 1, as this type of pattern is more likely removed by the auto-encoder, and given the constraint shown that the same prefix must not be in the same list twice as discussed in Section 3.2, the most effective $G$ for BitMark are $G = \{01, 10\}$ and $G = \{00, 11\}$. We choose $G = \{01, 10\}$ throughout all experiments.

**Changing the Length of the Bit Sequence.** We additionally ablate the size of the bit sequence in Table 7. For each sequence size $n$, if both lists should have the same size, we choose $\frac{2^n}{2}$ out of $2^n$ possibilities without replacement, so there are $\binom{2^n}{2^{n-1}}$ possibilities of constructing the $G$ and $R$. For the 2-bit sequences, flipping the green and red list has only limited impact on the final z-score, thus for the 3-bit sequences we only analyze all unique choices, *i.e.,* we do not interchange green and red list, leaving us with 35 distinct possible green and red lists. For $n = 3$, two $G$ have a similar $z$ compared to the best 2-bit sequences. These choices have the same properties as discussed above, namely (1) they equally often bias towards 0 and 1 and (2) none of the sequences share the same prefix, *i.e.,* the biasing is always effective.

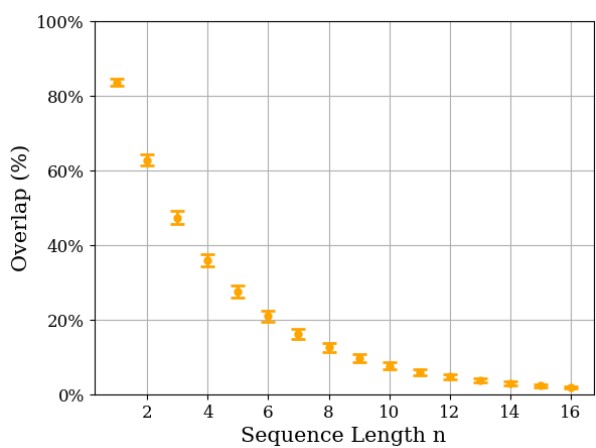

Figure 3: **Longer sequences are less consistent after re-encoding.** We analyze the consistency of sequences of length $n$ after re-encoding over the whole image, following the setup of Figure 1.

## E  Experimental Setup

### E.1  Hyperparameters

For the experiments on radioactivity, we chose 1,000 prompts from the MS-COCO 2014 validation dataset, generated the respective number of images with the $M_1$ Infinity model and then fine-tuned the second model $M_2$ for 5 epochs with a learning rate of $10^{-4}$ and a batch-size of 4 (batch-size of 1 for Infinity-2B).

Since VAR [35] and RAR [41] are class-conditional models for the ImageNet1k dataset, we first generated ImageNet fakes using Infinity-2B with the ImagenNet class name and "A photo of {class name}" as prompt. We used the same hyperparameters such as batch-size, number of epochs and learning rate consistent with the experiments on Infinity-2B and Stable Diffusion 2.1.

### E.2  Environment

Our experiments are performed on Ubuntu 22.04, with Intel(R) Xeon(R) Gold 6330 CPU and NVIDIA A40 Graphics Card with 40 GB of memory.

Table 7: **Green and Red list selection.** We analyze all possible green and red lists for $n = 1$, $n = 2$ and $n = 3$ with $\delta = 2$ and 1,000 images. We also report the fraction of 1's that are in the bit representation during generation of the image (**1s Gen**) and after re-encoding the image using the image encoder (**1s Enc**).

| Green List $G$ | Red List $R$ | Average Z-Score | 1s Gen | 1s Enc |
|---|---|---|---|---|
| $\{0\}$ | $\{1\}$ | 6.18 | 31.50% | 49.47% |
| $\{1\}$ | $\{0\}$ | 2.39 | 64.52% | 50.21% |
| $\{11, 00\}$ | $\{01, 10\}$ | 91.35 | 49.74% | 49.98% |
| $\{01, 10\}$ | $\{00, 11\}$ | 90.33 | 49.97% | 49.99% |
| $\{00, 10\}$ | $\{01, 11\}$ | 6.52 | 31.85% | 49.44% |
| $\{11, 01\}$ | $\{00, 10\}$ | 2.56 | 63.70% | 50.22% |
| $\{00, 01\}$ | $\{10, 11\}$ | 0.25 | 49.92% | 49.98% |
| $\{11, 10\}$ | $\{00, 01\}$ | -0.25 | 49.92% | 49.98% |
| $\{000, 011, 100, 111\}$ | $\{001, 010, 101, 110\}$ | 88.00 | 49.75% | 49.98% |
| $\{000, 011, 110, 111\}$ | $\{001, 010, 100, 101\}$ | 87.91 | 49.75% | 49.98% |
| $\{000, 011, 101, 111\}$ | $\{001, 010, 100, 110\}$ | 67.90 | 49.75% | 49.98% |
| $\{000, 100, 110, 111\}$ | $\{001, 010, 011, 101\}$ | 44.95 | 40.92% | 49.52% |
| $\{000, 001, 011, 111\}$ | $\{010, 100, 101, 110\}$ | 32.91 | 58.19% | 50.24% |
| $\{000, 001, 010, 101\}$ | $\{011, 100, 110, 111\}$ | 32.85 | 44.09% | 49.68% |
| $\{000, 010, 011, 111\}$ | $\{001, 100, 101, 110\}$ | 29.04 | 40.92% | 49.52% |
| $\{000, 100, 101, 111\}$ | $\{001, 010, 011, 110\}$ | 29.04 | 40.92% | 49.52% |
| $\{000, 101, 110, 111\}$ | $\{001, 010, 011, 100\}$ | 26.43 | 40.92% | 49.52% |
| $\{000, 011, 100, 110\}$ | $\{001, 010, 101, 111\}$ | 20.11 | 49.75% | 49.98% |
| $\{000, 001, 011, 110\}$ | $\{010, 100, 101, 111\}$ | 10.03 | 58.19% | 50.24% |
| $\{000, 001, 011, 100\}$ | $\{010, 101, 110, 111\}$ | 8.72 | 58.19% | 50.24% |
| $\{000, 001, 010, 100\}$ | $\{011, 101, 110, 111\}$ | 6.48 | 44.09% | 49.68% |
| $\{000, 010, 100, 101\}$ | $\{001, 011, 110, 111\}$ | 6.20 | 33.07% | 49.48% |
| $\{000, 010, 100, 110\}$ | $\{001, 011, 101, 111\}$ | 5.99 | 33.07% | 49.48% |
| $\{000, 010, 011, 100\}$ | $\{001, 101, 110, 111\}$ | 5.55 | 40.92% | 49.52% |
| $\{000, 001, 010, 110\}$ | $\{011, 100, 101, 111\}$ | 3.74 | 44.09% | 49.68% |
| $\{000, 010, 101, 110\}$ | $\{001, 011, 100, 111\}$ | 3.21 | 33.07% | 49.48% |
| $\{000, 010, 011, 110\}$ | $\{001, 100, 101, 111\}$ | 2.93 | 40.92% | 49.52% |
| $\{000, 100, 101, 110\}$ | $\{001, 010, 011, 111\}$ | 2.93 | 40.92% | 49.52% |
| $\{000, 010, 100, 111\}$ | $\{001, 011, 101, 110\}$ | 2.14 | 33.07% | 49.48% |
| $\{000, 001, 100, 101\}$ | $\{010, 011, 110, 111\}$ | 0.25 | 49.92% | 49.98% |
| $\{000, 001, 010, 011\}$ | $\{100, 101, 110, 111\}$ | 0.25 | 49.92% | 49.98% |
| $\{000, 011, 100, 101\}$ | $\{001, 010, 110, 111\}$ | 0.11 | 49.75% | 49.98% |
| $\{000, 001, 101, 110\}$ | $\{010, 011, 100, 111\}$ | 0.07 | 49.92% | 49.98% |
| $\{000, 011, 101, 110\}$ | $\{001, 010, 100, 111\}$ | 0.02 | 49.75% | 49.98% |
| $\{000, 001, 100, 110\}$ | $\{010, 011, 101, 111\}$ | -0.05 | 49.92% | 49.98% |
| $\{000, 001, 101, 111\}$ | $\{010, 011, 100, 110\}$ | -0.26 | 49.92% | 49.98% |
| $\{000, 001, 100, 111\}$ | $\{010, 011, 101, 110\}$ | -0.37 | 49.92% | 49.98% |
| $\{000, 001, 110, 111\}$ | $\{010, 011, 100, 101\}$ | -0.55 | 49.92% | 49.98% |
| $\{000, 010, 101, 111\}$ | $\{001, 011, 100, 110\}$ | -0.65 | 33.07% | 49.48% |
| $\{000, 010, 110, 111\}$ | $\{001, 011, 100, 101\}$ | -0.85 | 33.07% | 49.48% |
| $\{000, 001, 011, 101\}$ | $\{010, 100, 110, 111\}$ | -1.45 | 58.19% | 50.24% |
| $\{000, 010, 011, 101\}$ | $\{001, 100, 110, 111\}$ | -12.98 | 40.92% | 49.52% |
| $\{000, 001, 010, 111\}$ | $\{011, 100, 101, 110\}$ | -13.39 | 44.09% | 49.68% |

## F  Additional Experiments

In this Section we ablate the behavior of BitMark on the Infinity-2B model.

### F.1  Scalewise Analysis of BitMark

Next, we analyze the scalewise impact of BitMark during image generation. Table 8 depicts information about the generation of images with Infinity-2B, averaged over 10,000 images. It shows that the entropy is low in earlier scales, meaning the bits are chosen with high certainty as these depend more on the input prompt, but large on later scales, which means sampling either of the possible values for the bit can lead to different, but both high-quality content.

Table 8: **Statistics regarding the infinity generation process for 10,000 images.** Entropy, Green Fraction& Re-Encoding Loss are relative values. The Re-Encoding Loss displays the number of bits changed during re-encoding. We report the mean (std) for all values.

| Scale | Entropy | $z$ | Green Fraction | Re-Encoding Loss | Number of Tokens (Bits) |
|---|---|---|---|---|---|
| 1 | 0.051 (0.037) | 0.01 (0.876) | 0.501 (0.079) | 0.068 (0.063) | 1 (32) |
| 2 | 0.108 (0.044) | -0.195 (1.114) | 0.491 (0.049) | 0.132 (0.063) | 4 (128) |
| 3 | 0.133 (0.034) | -0.463 (1.226) | 0.49 (0.027) | 0.145 (0.041) | 16 (512) |
| 4 | 0.178 (0.029) | -0.522 (1.217) | 0.492 (0.018) | 0.2 (0.031) | 36 (1,152) |
| 5 | 0.202 (0.024) | -0.558 (1.199) | 0.494 (0.013) | 0.226 (0.024) | 64 (2,048) |
| 6 | 0.215 (0.021) | -0.663 (1.229) | 0.495 (0.009) | 0.247 (0.02) | 144 (4,608) |
| 7 | 0.223 (0.019) | -0.5 (1.266) | 0.497 (0.007) | 0.239 (0.017) | 256 (8,192) |
| 8 | 0.223 (0.015) | -0.349 (1.187) | 0.498 (0.005) | 0.277 (0.016) | 400 (12,800) |
| 9 | 0.226 (0.014) | -0.122 (1.173) | 0.5 (0.004) | 0.279 (0.014) | 576 (18,432) |
| 10 | 0.243 (0.014) | 0.157 (1.165) | 0.5 (0.003) | 0.264 (0.012) | 1,024 (32,768) |
| 11 | 0.237 (0.012) | 0.227 (1.161) | 0.501 (0.003) | 0.319 (0.011) | 1,600 (51,200) |
| 12 | 0.233 (0.012) | 0.329 (1.198) | 0.501 (0.002) | 0.328 (0.01) | 2,304 (73,728) |
| 13 | 0.234 (0.014) | 0.462 (1.173) | 0.501 (0.002) | 0.212 (0.012) | 4,096 (131,072) |
| 1-13 | 0.193 (0.012) | 0.232 (1.644) | 0.497 (0.008) | 0.266 (0.009) | 10,521 (336,372) |

Table 9: **Statistics regarding the infinity generation process with BitMark applied on all scales for 10,000 images,** $\delta = 2$**.** Entropy, Green Fraction, Re-Encoding Loss & Changed Bits are relative values. The Re-Encoding Loss displays the number of bits changed during re-encoding, whereas Changed Bits display the number of bits changed during the generation process. We report the mean (std) for all values.

| Scale | Entropy | $z$ | Green Fraction | Re-Encoding Loss | Changed Bits |
|---|---|---|---|---|---|
| 1 | 0.051 (0.037) | 0.218 (0.897) | 0.52 (0.081) | 0.088 (0.076) | 0.09 (0.077) |
| 2 | 0.108 (0.043) | 0.854 (1.105) | 0.538 (0.049) | 0.181 (0.079) | 0.179 (0.075) |
| 3 | 0.132 (0.032) | 2.37 (1.452) | 0.552 (0.032) | 0.2 (0.053) | 0.208 (0.05) |
| 4 | 0.175 (0.028) | 3.965 (1.613) | 0.558 (0.024) | 0.26 (0.037) | 0.279 (0.042) |
| 5 | 0.2 (0.024) | 6.252 (1.812) | 0.569 (0.02) | 0.286 (0.029) | 0.313 (0.035) |
| 6 | 0.213 (0.021) | 10.118 (2.304) | 0.575 (0.017) | 0.305 (0.022) | 0.329 (0.029) |
| 7 | 0.224 (0.019) | 15.701 (2.832) | 0.587 (0.016) | 0.3 (0.02) | 0.346 (0.027) |
| 8 | 0.224 (0.016) | 16.247 (2.788) | 0.572 (0.012) | 0.329 (0.016) | 0.345 (0.022) |
| 9 | 0.228 (0.015) | 20.876 (3.341) | 0.577 (0.012) | 0.333 (0.013) | 0.351 (0.02) |
| 10 | 0.244 (0.015) | 32.837 (5.104) | 0.591 (0.014) | 0.319 (0.011) | 0.373 (0.019) |
| 11 | 0.237 (0.013) | 29.535 (5.06) | 0.565 (0.011) | 0.361 (0.009) | 0.363 (0.017) |
| 12 | 0.242 (0.013) | 34.973 (6.681) | 0.564 (0.012) | 0.37 (0.008) | 0.368 (0.017) |
| 13 | 0.247 (0.014) | 64.333 (11.823) | 0.589 (0.016) | 0.256 (0.015) | 0.374 (0.019) |
| 1-13 | 0.194 (0.012) | 90.784 (14.852) | 0.566 (0.013) | 0.312 (0.009) | 0.366 (0.017) |

Table 9, depicts using BitMark with $\delta = 2$. Applying BitMark changes up to 37% of bits per scale, yet the re-encoding loss only increases by 5% compared to the non watermarked images. Table 10 further shows the correlation between different analyzed factors. As expected, there is a high correlation between changed bits and entropy, as due to the soft-biasing we mostly change bits of low entropy.

Table 10: **Correlations of different factors when applying BitMark on all scales within the Infinity generation process.** Entropy, Green Fraction, Changed Bits & Re-Encoding Loss are relative values. The Re-Encoding Loss displays the number of bits changed during re-encoding, whereas Changed Bits display the number of bits changed during the generation process. 10,000 images, $\delta = 2$.

|  | Entropy | Changed Bits | $z$ | Green Fraction |
|---|---|---|---|---|
| Changed Bits | 0.999 | - | - | - |
| $z$ | 0.706 | 0.689 | - | - |
| Green Fraction | 0.918 | 0.917 | 0.642 | - |
| Re-Encoding Loss | 0.926 | 0.931 | 0.475 | 0.751 |

We further observe that the model produces high-level information in the image during the initial scales, forming a coarse shape of the main visual components, such as the object and the background (see Figure 5. In larger scales, the model predicts the details of the image (see Figure 6). As shown in Table 11, earlier and late scales contribute differently to the robustness of BitMark. Earlier scales are more robust against like CtrlRegen+, as CtrlRegen+ does not change the content of the image itself, but mostly high level features.

Table 11: **Robustness of BitMark applied to different scales with $\delta = 2$.** We report the TPR@1%FPR (%) for the different attacks on 5,000 watermarked images.

| Scales | Conventional Attacks | | | | | | | Reconstruction Attacks | |
|---|---|---|---|---|---|---|---|---|---|
|  | None | Noise | Blur | Color | Crop | Rotation | JPEG | SD2.1-VAE | CtrlRegen+ |
| 1-10 | 100.0 | 97.9 | 98.7 | 97.6 | 61.6 | 15.0 | 100.0 | 100.0 | 82.1 |
| 11-13 | 100.0 | 96.7 | 98.6 | 99.5 | 20.0 | 9.8 | 100.0 | 100.0 | 15.5 |
| 1-13 | 100.0 | 99.6 | 99.9 | 99.8 | 98.8 | 20.1 | 100.0 | 100.0 | 91.6 |

## F.2 Robustness

Table 12 shows an ablation of the robustness of BitMark given different values of $\delta$. BitMark achieves a high TPR@1%FPR against most attacks even for a low watermarking strength $\delta = 1$. While stronger robustness can be achieved with higher $\delta$ values, there is a direct tradeoff with the image quality, which we show qualitatively in Figure 7.

Table 12: **BitMark is robust against watermark removal attacks.** We report the TPR@1%FPR (%) for the different attacks and baselines.

| $\delta$ | Conventional Attacks | | | | | | | Reconstruction Attacks | |
|---|---|---|---|---|---|---|---|---|---|
|  | None | Noise | Blur | Color | Crop | Rotation | JPEG | SD2.1-VAE | CtrlRegen+ |
| 1 | 100.0 | 97.8 | 99.2 | 99.3 | 68.5 | 14.5 | 100.0 | 100.0 | 61.9 |
| 2 | 100.0 | 99.6 | 99.9 | 99.8 | 98.8 | 20.1 | 100.0 | 100.0 | 91.6 |
| 3 | 100.0 | 99.7 | 99.9 | 99.9 | 99.8 | 23.8 | 100.0 | 100.0 | 97.0 |
| 4 | 100.0 | 99.9 | 99.9 | 100.0 | 99.9 | 29.2 | 100.0 | 100.0 | 99.0 |
| 5 | 100.0 | 100.0 | 100.0 | 99.9 | 100.0 | 34.3 | 100.0 | 100.0 | 99.8 |

**Rotation.** We ablate the robustness of BitMark against different degrees of rotation and different watermarking strengths $\delta$. We find that with $\delta = 2$ BitMark is robust for rotation degrees $\pm 30$. Additionally we find an increase in TPR@1%FPR for a rotation of 180°.

We further show the improved detection performance leveraging a rotation estimation tool [21] in Table 13. After re-rotating the image, the detection performance increases from TPR@1%FPR=0.184 to TPR@1%FPR=0.983. In addition, we consider an even harder case: after the random rotation, we apply a central crop to the image with 50% cropping ratio, such that all the black corners during rotation are completely removed. We show the results in Table 14.

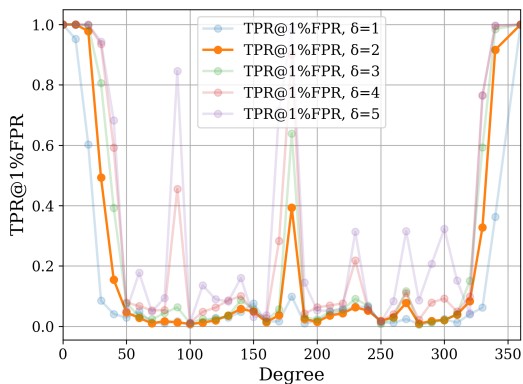

Figure 4: **BitMark is robust against rotation within** $\pm 30$ **degrees.** The TPR@1%FPR for different $\delta$ and rotation degrees.

Table 13: **Impact of rotation correction on detection performance.** Images are randomly rotated between 0° and 360°.

| Setting | AUC (%) | Accuracy (%) | TPR@1%FPR (%) | Green Fraction (%) |
|---|---|---|---|---|
| No rotation correction | 60.4 | 60.8 | 18.4 | 50.2 |
| After rotation correction | 99.5 | 99.2 | 98.3 | 52.0 |

Table 14: **Impact of rotation correction and cropping on detection performance.** Images are randomly rotated between 0° and 360° with 50% center crop.

| Setting | AUC (%) | Accuracy (%) | TPR@1%FPR (%) | Green Fraction (%) |
|---|---|---|---|---|
| No rotation correction | 59.8 | 58.9 | 9.9 | 50.1 |
| After rotation correction | 89.1 | 86.5 | 66.3 | 50.5 |

### F.3 Image Quality

Table 15 shows that our chosen biases $\delta$ does not negatively impact the image quality for Infinity-2B ($\delta = 2$), Infinity-8B ($\delta = 1.5$) and Instella IAR ($\delta = 1.5$)

Additionally, we qualitatively analyze the impact of applying BitMark with different $\delta$ values. Figure 7 shows that applying BitMark with $\delta \leq 3$ keeps the semantic structure and quality of the image the same, while changing small features, *e.g.,* the color of the train in the first row. For larger watermarking strengths, the image quality starts to deteriorate, where artifacts become visible in the final image.

Table 15: **BitMark does not decrease the image quality.** Changes in FID, KID and CLIP relative to non-watermarked images.

| Model | $\Delta$FID | $\Delta$KID | $\Delta$CLIP |
|---|---|---|---|
| Infinity 2B, $\delta = 2$ | -2.1169 | -0.0016 | 0.1086 |
| Infinity 8B, $\delta = 1.5$ | 0.5299 | -0.0018 | 0.2792 |
| Instella IAR, $\delta = 1.5$ | 1.1471 | 0.0004 | -0.8361 |

### F.4 Timing

We present the average elapsed time (in seconds) for the generation of an image with Infinity, when the watermark is applied to all scales (resolutions) and when using the standard generation process (without a watermark). For hardware information refer to Section E.2. Table 16 shows that BitMark results in negligible overhead to image generation and allows for a fast detection in under 0.5 seconds.

Table 16: **Timing results for image processing and detection per image.** We report the mean (std) for 1,000 images.

|  | BitMark | No watermark | Detection |
|---|---|---|---|
| Seconds/Image | 2.5817 (0.0063) | 2.3196 (0.0043) | 0.4491 (0.0046) |

## F.5 Radioactivity to IARs

Table 17: **We analyze the radioactivity of BitMark for class-conditional autoregressive models.** We report the TPR@1%FPR (%).

| Type of $M_1$ | Type of $M_2$ | Output of $M_1$ | Output of $M_2$ |
|---|---|---|---|
| Infinity-2B | VAR-16 | 100.0 | 24.2 |
| Infinity-2B | VAR-20 | 100.0 | 25.8 |
| Infinity-2B | VAR-24 | 100.0 | 25.7 |
| Infinity-2B | VAR-30 | 100.0 | 25.6 |
| Infinity-2B | RAR-B | 100.0 | 4.3 |
| Infinity-2B | RAR-L | 100.0 | 3.3 |
| Infinity-2B | RAR-XL | 100.0 | 3.9 |
| Infinity-2B | RAR-XXL | 100.0 | 4.1 |

Table 17 reports the radioactivity of BitMark for the class-conditional autoregressive models, namely, **VAR** [35], which operates in the multi-scale manner similarly to Infinity, and **RAR** [41], which generates tokens only on a single scale in an autoregressive fashion. We find that BitMark exhibits detectable traces of radioactivity for the class-to-image autoregressive models models with TPR$>>$1%@FPR=1%, with slightly lower traces for RAR (probably due to its single scale generation) than VAR. A potential cause of the overall relatively lower radioactivity for VAR and RAR is stemming from the class-conditional setting, instead of the text-to-image setting as in Infinity. +

## F.6 Natural Images

To ensure the validity of our hypothesis we compute the average number of members of the green list and $z$ for natural images from different datasets, as well as images generated for the non-watermarked Infinity model. Table 18 shows that indeed, the green fraction for non-watermarked images is 50%.

Table 18: **Detection on non-watermarked samples.** Mean (std) of $z$ and green fractions for 1,000 non-watermarked samples.

| Dataset | $z$ | Green Fraction |
|---|---|---|
| Imagenet1k | 0.6311 (1.966) | 0.5005 (0.0017) |
| MS-COCO2014 | 0.6773 (1.735) | 0.5006 (0.0015) |
| Infinity-2B generated | 0.3620 (1.641) | 0.5003 (0.0014) |

We additionally show the FPR for different natural datasets (ImageNet-1k, LAION POP) and Infinity-2B generated images, given the 1%FPR threshold that we used for MS-COCO in Table 19.

Table 19: **False Positive Rate of choosing the MS-COCO 1%FPR threshold on different datasets.** FPR measured on N=1000 non-watermarked samples.

| Dataset | FPR@1%MS-COCO |
|---|---|
| MS-COCO | 0.01 |
| ImageNet-1k | 0.011 |
| LAION POP | 0.021 |
| Infinity-2B Generated | 0.0 |

# G Generalization to Diffusion

Additionally to the bitwise autoregressive image modeling that we explore in Section 4 we extend our BitMark to bitwise diffusion models, namely BiGR [12]. We employ two variations of BiGR, the L-256x256 with 24 bits per token and L-512x512 which leverages 32 bits per token. We set the Gumbel temperature to 0.01, the cfg to 2.5, the number of iterations as well as the diffusion timesteps to 10 each. We further notice that the detection performance for BiGR increases greatly if, before encoding, a slight amount of noise is added to the image.

In Table 20 we show that BitMark has negligible impact on the quality of the generated images for both BiGR model sizes. We use 5,000 MS-COCO prompts to generate both BiGR clean and watermarked data. On these images we then compute image quality metrics FID, KID and CLIP. The difference in FID and KID score shows that the watermarked images have lower FID and KID, a sign of more similarity to the MS-COCO baseline images. The CLIP score only increases slightly, showing only minor impact of BitMark to the prompt following capabilities of the model.

Additionally we show the robustness of BitMark in Table 21, it achieves significant TPR@1%FPR for all attacks. Especially against the advanced reconstruction attack BitMark achieves 75.2 and 69.2%TPR@1%FPR at resolution of $256 \times 256$ and $512 \times 512$, respectively. We note that, BiGR generates smaller images with only 256 or 1024 tokens, leading to a lower robustness compared to the Infinity-2B and 8B model.

Table 20: **BitMark does not decrease the image quality.** Changes in FID, KID and CLIP relative to non-watermarked images.

| Model | $\Delta$FID | $\Delta$KID | $\Delta$CLIP |
|---|---|---|---|
| BiGR 256x256, $\delta = 3$ | -3.0304 | -0.0020 | 0.0017 |
| BiGR 512x512, $\delta = 3$ | -3.6660 | -0.0024 | 0.1329 |

Table 21: **Watermark robustness against removal attacks.** We report the robustness (%TPR@1%FPR) for the different attacks. For BiGR, we added a slight amount of noise to all images before encoding.

| Watermark | Conventional Attacks | | | | | | | | | Reconstruction Attacks | |
|---|---|---|---|---|---|---|---|---|---|---|---|
| | None | Noise | Blur | Color | Rotate | Crop | JPEG | Vertical | Horizontal | VAE | CtrlRegen+ |
| BiGR $256 \times 256$ ($\delta = 3$) | 97.0 | 54.3 | 94.2 | 16.1 | 55.3 | 49.7 | 95.1 | 61.3 | 47.8 | 95.5 | 75.2 |
| BiGR $512 \times 512$ ($\delta = 3$) | 99.3 | 57.4 | 96.4 | 34.2 | 65.1 | 69.4 | 97.6 | 75.4 | 77.2 | 98.2 | 69.2 |

# H Analysis of the Advanced Attacks

## H.1 Analysis for the Bit-Flipper Attack

We present the pseudo code of the novel Bit-Flipper attack for multi-scale models in Algorithm 5. To apply the attack, we first encode the image into bits (line 1). During deconstructing the watermarked image, we construct a new attacked image. We recursively iterate through all scales $K$, retrieve the bit representations of the respective scale (line 4) and count how many bits are part of the green list and the red list (line 5). Then we define the probability of flipping a bit based on the green fraction and the flip factor $\phi$. Next, we change the bits which are part of the green list based on the random probability $p$ (line 8, 9). We follow the standard encoding and decoding process, to iterate through all resolutions of the image (line 10-13). Finally, we return the decoded version of the attacked image.

Figure 9 shows different flipping strengths $\phi$ for $\delta = 2$ on 1,000 watermarked samples. The most effective $\phi$ is $\phi = 2.2$ with a TPR@1%FPR below 50% (see Table 22). The decrease in image quality for Bit-Flipper for different $\phi$ is shown in Figure 9, as well a more detailed analysis of $\phi = 2.2$ on BitMark with a $\delta = 2$ is displayed in Figure 10.

**Algorithm 5** Bit-Flipper Attack

---

**Inputs:** Image $im$, green list $G$, red list $R$, image decoder $\mathcal{D}$.
**Hyperparameters:** Flip factor $\phi$, steps $K$ (number of resolutions), resolutions $(h_i, w_i)_{i=1}^{K}$, the number of bits for resolution $i$ is $t_i$, $n$ - the length of the bit vector, $|s|$ the total number of bit sequences.
$e = \mathcal{E}(im)$
$img = \text{init}()$
**for** $i = 1, \ldots, K$ **do**
$\quad u_i = \mathcal{Q}(\text{Interpolate}(e, h_i, w_i))$
$\quad C_i = \text{Count}((b_1, \ldots, b_{t_i}), G)$
$\quad p = (\frac{C_i}{|s_i|} - 0.5) * \phi$
$\quad$**for** $j = n, \ldots, t_i$ **do**
$\quad\quad$**if** $(b_{j-n}, \ldots, b_j) \in G \land random() < p$ **then**
$\quad\quad\quad b_j = \neg b_j$ {Flip bit with probability $p$ if sequence is in green list}
$\quad z_i = \text{Lookup}(u_i)$
$\quad z_i = \text{Interpolate}(z_i, h_K, w_K)$
$\quad e = e - \phi_i(z_i)$
$\quad img = img + \phi_i(z_i)$
**Return:** attacked image $\mathcal{D}(img)$

---

Table 22: **Applying Bit-Flipper with different factors $\phi$ on watermarked images with $\delta = 2$.** We report the TPR@1%FPR for 1,000 Images.

| $\phi$ | TPR@1%FPR (%) | $z$ |
|---|---|---|
| 1.8 | 0.878 | 10.06 (4.24) |
| 2.0 | 0.845 | 9.17 (4.16) |
| 2.2 | 0.472 | 5.52 (5.07) |
| 2.4 | 0.595 | 8.06 (7.28) |
| 2.6 | 0.52 | 6.97 (6.96) |

## H.2 Watermarks in the Sand Attack

As shown by the qualitative analysis in Figure 8, the watermarks in the sand attack causes a significant loss of details and produces many perceptible artifacts, limiting the applicability of this attack.

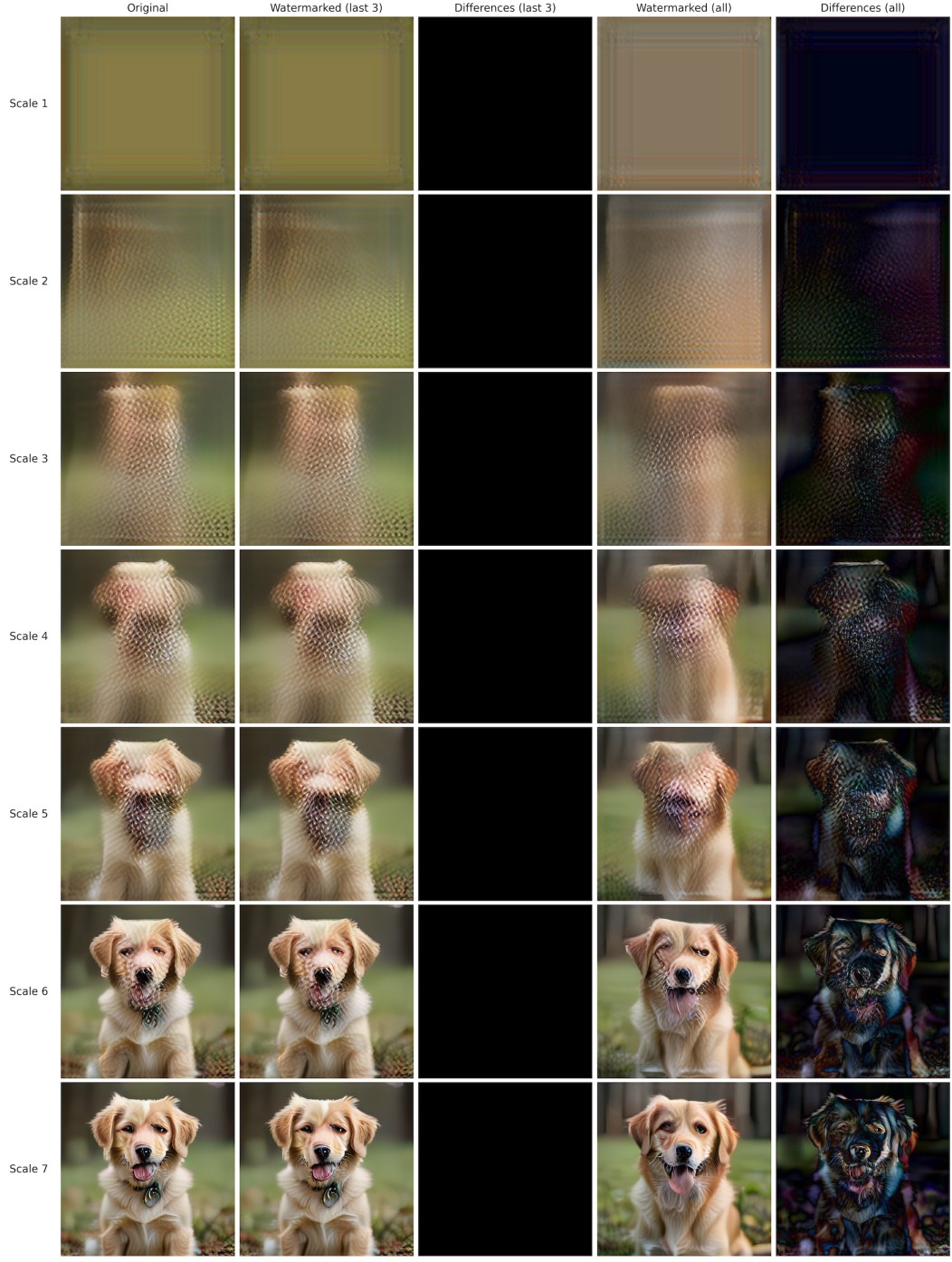

Figure 5: **Visualization of different images on different scales.** The image on scale $i$ refers to the cumulative image obtained by adding up 1 to $i$ scales. We visualize five sets of images: *1) Original* images without watermarking, *2) Watermarked (last 3)*, where the watermarks are generated on scale 11 to 13, *3) Differences (last 3)*, which is the difference between the original and watermarked images (last 3), *4) Watermarked (all)*, where the watermarks are generated on all scales, *5) Differences (all)*, which is the difference between the original and watermarked images (all). We visualize scales 1-7.

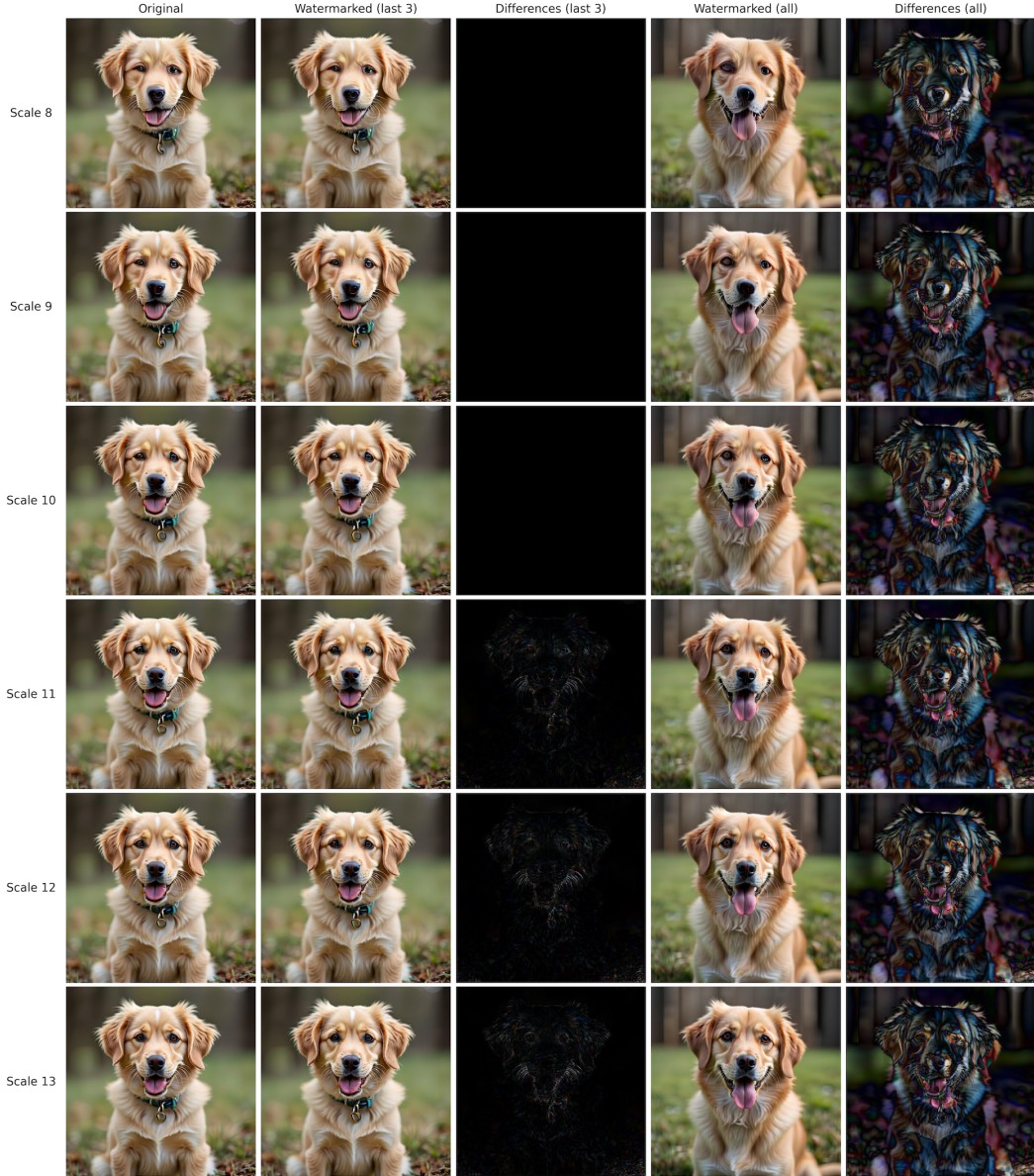

Figure 6: **Visualization of different images on different scales.** The image on scale $i$ refers to the cumulative image obtained by adding up 1 to $i$ scales. We visualize five sets of images: *1) Original* images without watermarking, *2) Watermarked (last 3)*, where the watermarks are generated on scale 11 to 13, *3) Differences (last 3)*, which is the difference between the original and watermarked images (last 3), *4) Watermarked (all)*, where the watermarks are generated on all scales, *5) Differences (all)*, which is the difference between the original and watermarked images (all). We visualize scales 8-13.

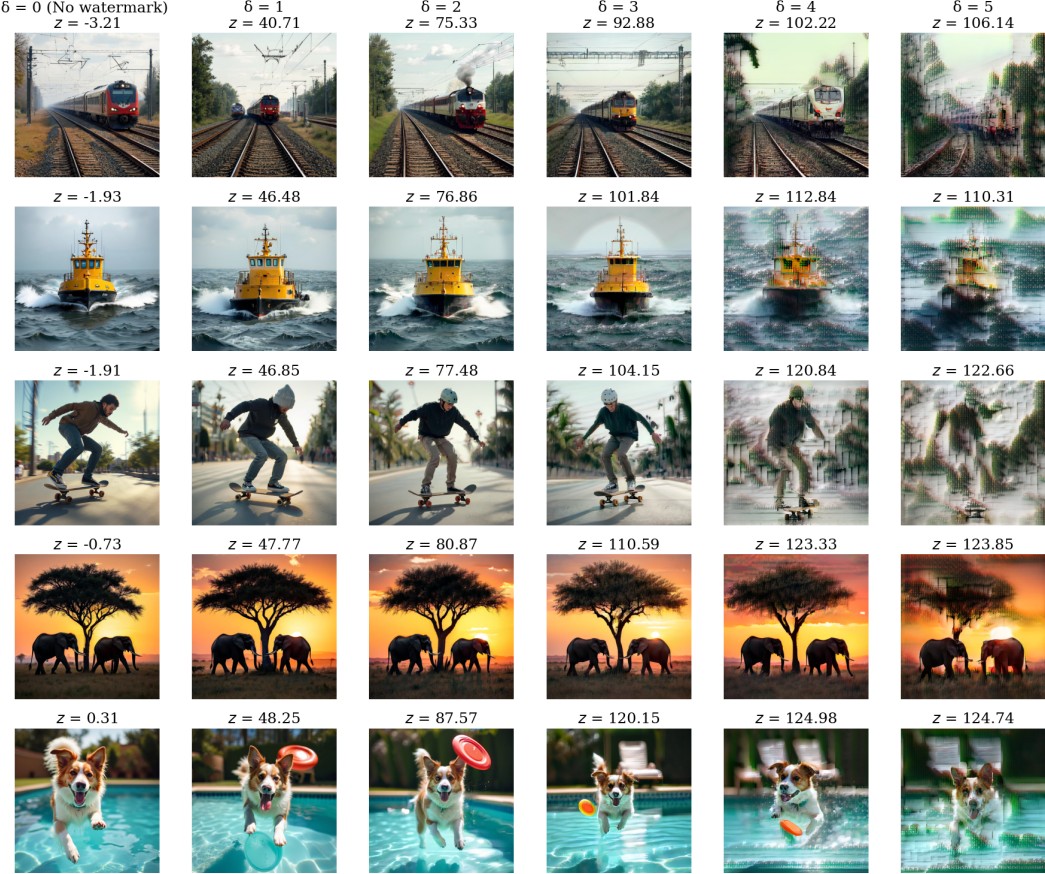

Figure 7: **Artifacts start to form in high detailed images already with $\delta = 3$.** Qualitative analysis of the impact of the choice of watermarking strength ($\delta$) on the perceived image quality.

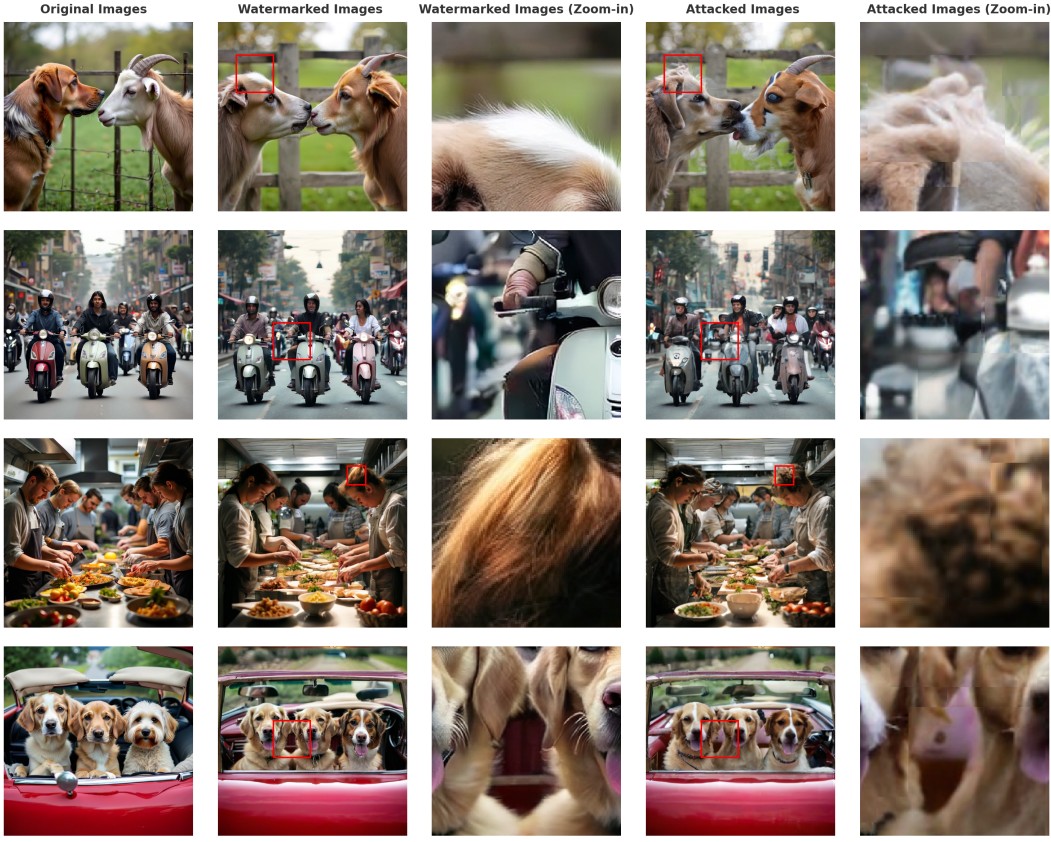

Figure 8: **Watermarks in the Sand has non-negligible impact on image quality.** Qualitative analysis of the impact of the watermarks in the sand. Here, we zoom in on some regions in the image to demonstrate the quality loss in details.

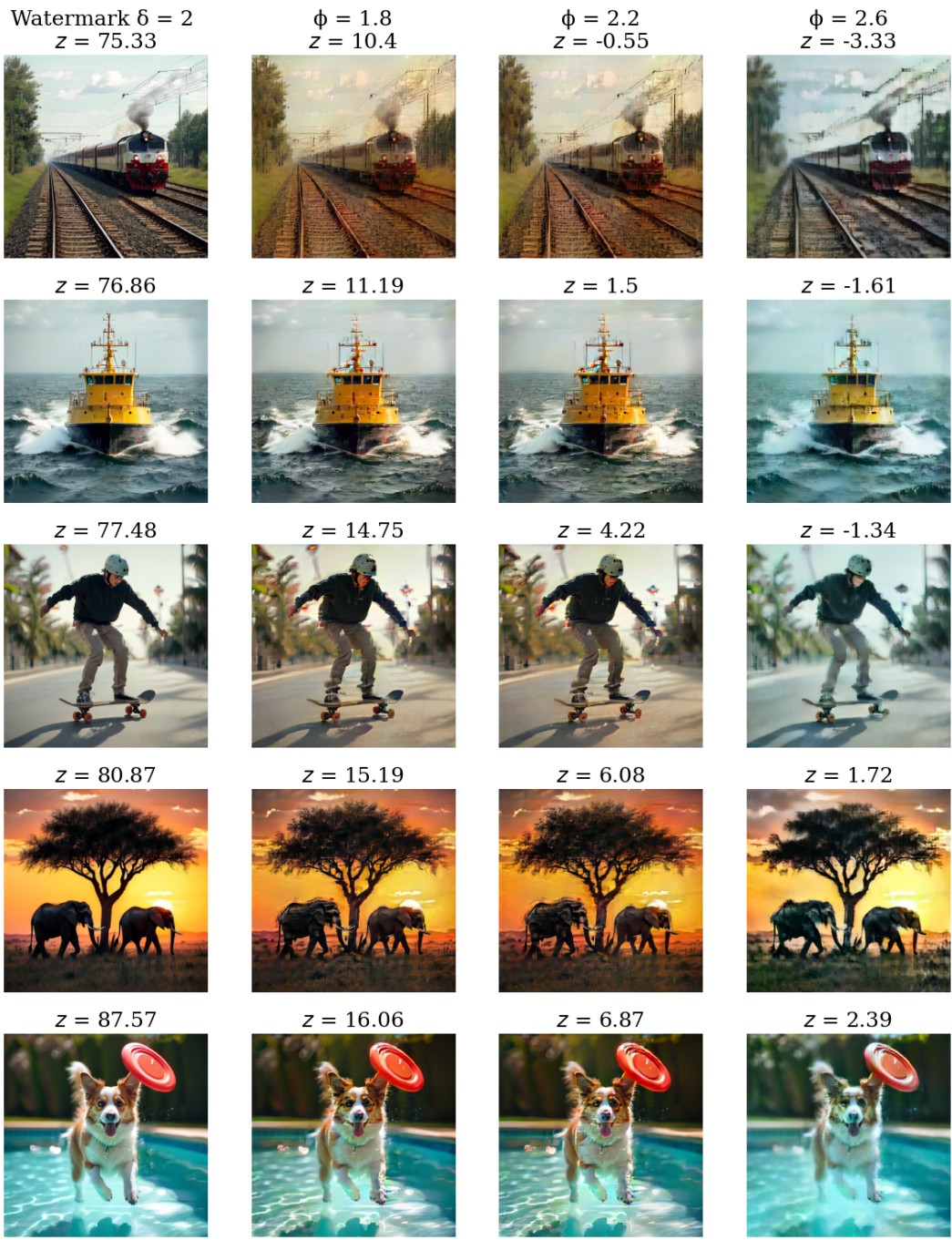

Figure 9: **A higher flip factor leads to worse image quality but better watermark removal.** Qualitative analysis of the impact of the Bit-Flipper with different $\phi$.

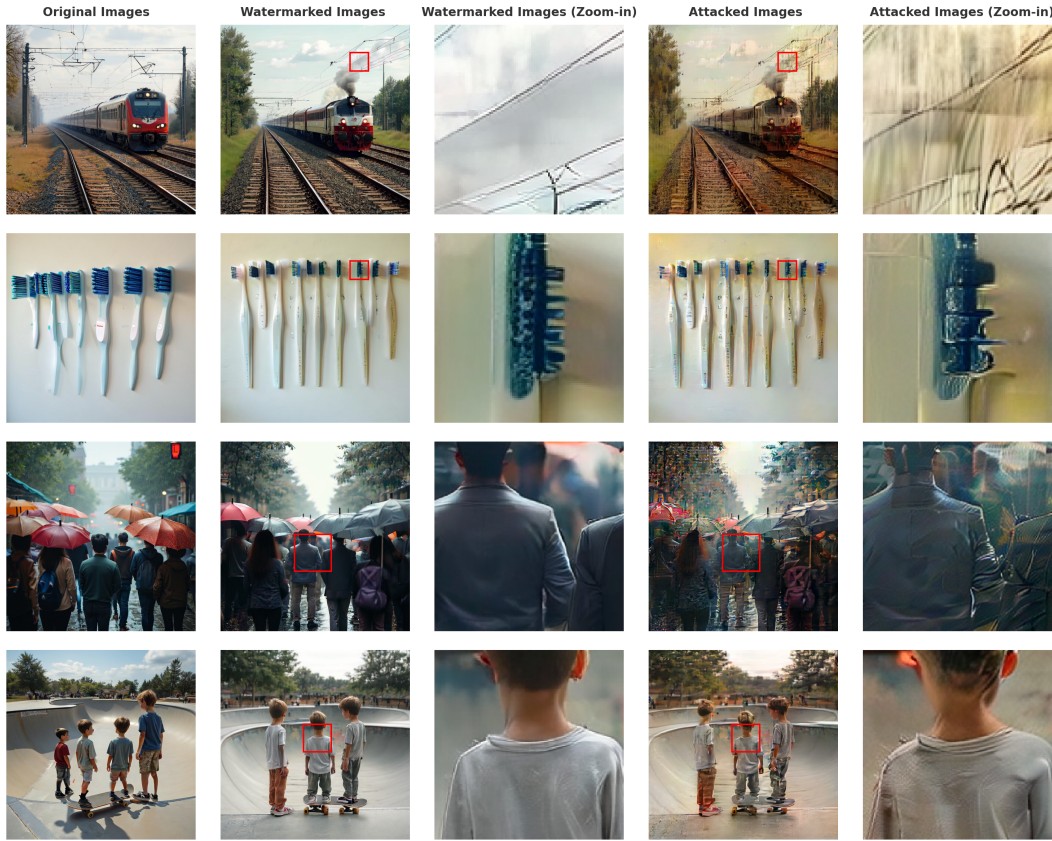

Figure 10: **The Bit-Flipper has strong impact on the image quality.** Here, we zoom in on some regions in the image to demonstrate the quality loss in details. $\phi = 2.2$, $\delta = 2$.

