# OpenReview forum: "BitMark: Watermarking Bitwise Autoregressive Image Generative Models"
_NeurIPS.cc/2025/Conference — NeurIPS 2025 poster_

### Official Review · Reviewer_NhKn · 2025-06-10

**Clarity:** 3
**Significance:** 2
**Originality:** 2
**Rating:** 4
**Confidence:** 4

**Summary:**

This paper introduces BitMark, a bit-level watermarking framework designed for Infinity, a state-of-the-art bitwise autoregressive image generation model. The core idea is to embed watermarks directly in the bit-level token stream during inference, aligning with Infinity’s discrete and hierarchical generation structure. BitMark uses green and red bit sequence lists to subtly bias generation toward detectable patterns with negligible visual distortion. The watermark is detectable via statistical testing on the bit distribution of re-encoded images and does not require model retraining. Key contributions include (1) a bit-level watermarking method aligned with Infinity’s architecture, (2) robustness to common and dedicated removal attacks (like CtrlRegen and Watermark in the Sand), (3) negligible impact on generation quality and speed, and (4) strong radioactivity: fine-tuned models on BitMark outputs also inherit the watermark, making it useful for provenance tracking and preventing model collapse. Extensive experiments on MS-COCO show strong detection accuracy, minimal perceptual impact for low watermark strength, and robustness to both synthetic and adversarial attacks.

**Questions:**

- How well does BitMark generalize to other bitwise autoregressive models beyond Infinity? Is it easily portable, or tightly coupled to specific implementation details?

- For the statistical detection test, how sensitive is the threshold to dataset shifts (e.g., web images vs. MS-COCO)? Can the test fail if the distribution of bit patterns in clean images varies? It would be good if you can provide empirical checks of FPR on domain-shifted datasets.

- The paper claims radioactivity even for Stable Diffusion models trained on BitMark images. This is previously considered non-trivial, and as far as I know, previous works [1] even specifically designed loss functions to increase the learnability of the watermarks. So I really wonder why is your method radioactive in nature.

[1]: Li et al. Towards Reliable Verification of Unauthorized Data Usage in Personalized Text-to-Image Diffusion Models. S&P 2025.

**Ethical Concerns:**

["NO or VERY MINOR ethics concerns only"]

**Final Justification:**

The authors rebuttal have addressed most of my concerns. So I raise my score to 4. The reason for not higher score is the modest novelty (as the method can be regarded as an extention of existinng LLM watermarks to autoregressive image generative models) and the mechanism under its radioactivity is not very theoretically well-justified. However, considering that the work is one of the earliest to explore watermarks for AR models, I would still support acceptance. The authors might explore it deeper in the future.

**Limitations:**

Yes.

**Quality:**

3

**Strengths And Weaknesses:**

## Strengths:

- The problem of watermarking VAR models and preventing model collapse from training on synthetic content is real and increasingly urgent. Thus the topic of this paper is important and timely.

- BitMark is the first watermarking technique tailored for bitwise autoregressive image generation models like Infinity.

- The watermark is shown to be robust across a large variety of attack vectors including conventional image transformations, advanced watermark-removal attacks (CtrlRegen, Watermark in the Sand), and even an adaptive bit-flipping attack crafted with white-box access.

- The watermark is shown to be highly radioactive even across model families (Infinity to Stable Diffusion), which is non-trivial and potentially useful for tracing data lineage.


## Weaknesses:

– The core technique, while adapted to a new domain, is heavily inspired by token-level watermarking schemes from LLMs (e.g., KGW, Unigram Watermark). The main contribution is repackaging the method for the bit-level of a specific image generation architecture. This raises questions about whether it’s truly innovative or just porting an old idea. Besides, the proposed watermarking technique seems to be specifically tailored for the Infinity model, as shown by its technical implementation and experimental validation. How would the proposed method extend to other VAR text to image models remains unknown, which would largely restrict the impact and scope of the paper.

- Despite claims of “negligible” image quality impact, Table 1 shows that at watermark strength 5 the FID degrades significantly and CLIP score drops by over 5 points. This shows that the watermark is not as negligible as claimed when pushed to be stronger.

- The radioactivity seems to be a strong and non-trivial property of BitMark, and it even transfers to Diffusion-based models (SD v2.1 as in this paper). However, as the authors said, previous watermarking methods designed for diffusion models do not exhibit strong radioactivity. The author attributes this radioactivity to " slightly changing all parts of a given generated image, including low-level and high-level features, which are then also partly adapted by M2.", however, I believe previous watermarks like tree-ring also satisifies this property. Thus  I am not convinced by this explanation and I believe more concrete analysis should be given to provide insights into the community.

---

> ### Author Rebuttal · Authors · 2025-07-31
>
> We thank the Reviewer for the detailed analysis of our paper. We appreciate that the Reviewer finds the topic of our paper important and timely. We are also glad that our comprehensive experiments clearly demonstrate the strengths of our watermark, such as the **non-trivial property of radioactivity**.
>
> In summary, **BitMark generalizes** to two additional models, we emphasize that BitMark has **negligible impact on the quality of generated outputs**, and we also empirically **analyzed the FPR on domain-shifted datasets**. We provide the answers in-line below:
>
> >** The core technique, while adapted to a new domain, is heavily inspired by token-level watermarking schemes from LLMs (e.g., KGW, Unigram Watermark). The main contribution is repackaging the method for the bit-level of a specific image generation architecture. This raises questions about whether it’s truly innovative or just porting an old idea. **
>
> Recent watermarking papers [1,2] focus on the token level, where the token overlap between generated and re-encoded images is small. This necessitates retraining of the encoder and decoder. Our key contribution is to demonstrate that operating at the lower level of bits offers a great advantage. Specifically, we demonstrate that the bit-level overlap between generated and re-encoded images is significant and sufficient to enable direct bit-level watermarking, without retraining the encoder and decoder. This allows us to provide the strongest known watermark for image autoregressive models and, as stated by Reviewer 9clc: *“the green-red list mechanism adapted from LLM watermarking is cleverly adapted to image generation scenario, with modifications to handle bitwise autoregression.”*
>
> >**How well does BitMark generalize to other bitwise autoregressive models beyond Infinity? Is it easily portable, or tightly coupled to specific implementation details?**
>
> We ran additional experiments to demonstrate that BitMark is applicable to two other bitwise models, namely BiGR [3] and Instella [4].  *BiGR* is an image generative model that harnesses binary latent codes and *Instella* is another text-to-image (T2I) Image AutoRegressive (IAR) model leveraging bitwise next-patch prediction.
>
> We extended our experimental analysis to include the additional models and added the experiments to Appendix E. We also report the results below:
>
> We calculated a representative p-value for 5000 watermarked images by first calculating the respective mean z-score, then standardizing this mean z-score using the mean and standard deviation derived from z-scores of 5000 MS-COCO images. $\Delta$ denotes the image quality difference to non-watermarked generated images. FID, KID & CLIP are reported for 10000 images.
>
> | Model  | p-value of mean z-score | $\Delta$ FID | $\Delta$ KID | $\Delta$ CLIP |
> |:-:|:-:|:-:|:-:|:-:|
> | Infinity 2B, $\delta=2$ | 1.0e-594 | -2.1169 | -0.0016 | 0.1086 |
> | Infinity 8B, $\delta=1.5$ | 1.3e-273 | 0.5299 | -0.0018 | 0.2792 |
> | BiGR 256x256, d24, $\delta=3$ | 1.16e-23 | -3.0304 | -0.0020 | 0.0017 |
> | BiGR 512x512, d32, $\delta=3$ | 3.37e-31 | -3.6660 | -0.0024 | 0.1329 |
> | Instella IAR, $\delta=1.5$ | 2.27e-69 | 1.1471 | 0.0004 | -0.8361 |
>
> The results show that we can effectively embed our BitMark into the images generated from BiGR and Instella, without harm to the quality of output images (relatively small changes of the FID, KID, and CLIP scores). The extremely low p-values for the respective models demonstrate that we can confidently reject the $H_0$ (null hypothesis) and claim that our watermark was indeed embedded in the generated images. Thus, BitMark generalizes to multiple models and architectures.
> >**Despite claims of “negligible” image quality impact, Table 1 shows that at watermark strength 5 the FID degrades significantly and CLIP score drops by over 5 points. This shows that the watermark is not as negligible as claimed when pushed to be stronger.**
>
> BitMark strength of $\delta \le 3$ has a negligible impact on the image quality, as depicted in Table 1, while being sufficiently strong, as shown in Table 3. We agree that a $\delta=5$ degrades the image quality significantly. Therefore, we chose to run all experiments with $\delta=2$ which does not significantly degrade the FID or CLIP score of the model.
>
> >**For the statistical detection test, how sensitive is the threshold to dataset shifts (e.g., web images vs. MS-COCO)? Can the test fail if the distribution of bit patterns in clean images varies? It would be good if you can provide empirical checks of FPR on domain-shifted datasets.**
>
> We show that our selected $z$ threshold is robust to the dataset shifts in Table 7 in the Appendix.
>
> We additionally computed the FPR for different natural datasets (ImageNet-1k, LAION POP, and Infinity-2B Generated images) given the 1%FPR threshold that we also used for MS-COCO.
>
> | Dataset | FPR@1%MS-COCO-FPR (N=1000) |
> |:-:|:-:|
> | MS-COCO | 0.01 (used baseline) |
> | ImageNet-1k | 0.011 |
> | LAION POP | 0.021 |
> | Infinity-2B Generated | 0.0 |
>
> Our results show that the selected threshold is not sensitive to shifts in the natural data distribution across a variety of datasets. For every dataset we compute the FPR and find that it is around the 1% FPR of the MS-COCO dataset, showing the cross-distribution robustness of the chosen threshold.
>
> >**Can the test fail if the distribution of bit patterns in clean images varies?**
>
> The p-values for the natural images are significantly above the threshold of 1%. Thus, we never observe false positives (we never mark a natural images as watermarked).
>
> >**The radioactivity seems to be a strong and non-trivial property of BitMark, and it even transfers to Diffusion-based models (SD v2.1 as in this paper). However, as the authors said, previous watermarking methods designed for diffusion models do not exhibit strong radioactivity. The author attributes this radioactivity to " slightly changing all parts of a given generated image, including low-level and high-level features, which are then also partly adapted by M2.", however, I believe previous watermarks like tree-ring also satisfies this property. Thus I am not convinced by this explanation and I believe more concrete analysis should be given to provide insights into the community.**
>
> Previous work [5] (that we cite in our submission) demonstrated that TreeRing is not radioactive even when SD v2.1 is adapted on images generated from the same model and watermarked with TreeRing. We fully agree with the Reviewer that the radioactivity property is non-trivial. We observe that while TreeRing only influences specific frequencies (low-mid range) in the initial Gaussian noise of the generation process, our BitMark is applied throughout the full generation process and impacts all frequencies in the latent space (is applied to all the resolutions). BitMark thus influences all parts of the image: the basic color in the first scales, then coarse grained structures at intermediate scales, and high level details in the last scales. We believe that this wide range of the influence, both in terms of the sheer number of modified bits and the multi-scale application, makes the generated images radioactive.
>
>
> ---
> If the above responses address some of the Reviewer's concerns, we would greatly appreciate it if they could consider updating their rating to reflect this.
>
> **References:**
>
> [1] Tong et al. “Training-Free Watermarking for Autoregressive Image Generation”, 2025.
>
> [2]  Jovanović et al. “Watermarking Autoregressive Image Generation”, 2025.
>
> [3] Hao et al. “BiGR: Harnessing Binary Latent Codes for Image Generation and Improved Visual Representation Capabilities”, ICLR 2025.
>
> [4] Wang et al. “Instella-T2I: Pushing the Limits of 1D Discrete Latent Space Image Generation”, 2025.
>
> [5] Dubiński et al. “Are Watermarks For Diffusion Models Radioactive?”, 2025.

---

> > ### Comment · Reviewer_NhKn · 2025-08-03
> > **Thanks for the rebuttal.**
> >
> > The authors rebuttal have addressed most of my concerns. So I raise my score to 4. The reason for not higher score is the modest novelty (as the method can be regarded as an extention of existinng LLM watermarks to autoregressive image generative models) and the mechanism under its radioactivity is still not very theoretically well-justified. However, considering that the work is one of the earliest to explore watermarks for AR models, I would still support acceptance. The authors might explore it deeper in the future.

---

> > > ### Author Response · Authors · 2025-08-04
> > > **Thank you very much for your feedback and raising the score**
> > >
> > > Dear Reviewer NhKn,
> > >
> > > We would like to thank the Reviewer for the positive feedback and insightful comments. The paper improved as a result of your review. We also appreciate raising the score.
> > >
> > > Regarding the novelty, the biggest difference is that we operate on the bit-level instead of the token-level as in the watermarking of LLMs. The important difference between our bit-sequence watermarking and the token-level watermarking is as follows:
> > >
> > > An image is tokenized into a $n \times n$ token matrix. Each token in this matrix has $m$ bits, where each bit is like as a *channel* of the matrix. Our watermarking method operates on the bit sequence across the $m$ channels (in each matrix token), not on the spatial arrangement of the matrix tokens. This makes our watermark much more robust and radioactive. For example, when we apply horizontal flipping to the matrix, we change the order of the columns in the $n \times n$ matrix. Many watermarks are not robust to this form of attack. However, in BitMark, the order of the m-bit sequence in each token stays the same. Since the watermark is embedded in the m-bit sequences rather than the spatial relationships between the $n \times n$ token matrix, horizontal flipping largely preserves the watermark signal. The number of bits where we embed the watermark is also much larger than the number of tokens, which allows the watermark to be radioactive, since much more information is transferred to another model.
> > >
> > > We are grateful for the suggestions: we will definitely explore more the theoretical aspects of radioactivity.
> > >
> > > With kind regards,
> > >
> > > Authors of paper #2950 (BitMark)

---

### Official Review · Reviewer_cNxC · 2025-06-19

**Clarity:** 3
**Significance:** 3
**Originality:** 2
**Rating:** 4
**Confidence:** 2

**Summary:**

In this paper, the authors introduce BitMark, a bitwise watermark for the Infinity model. Unlike the token space, the authors find that bits are more robust when retokenizing the image. Therefore, they first randomly split all possible bit sequences into green and red lists. During sampling, the green bit sequences are biased to have higher generation probabilities. As a result, watermarked images will contain more green bit sequences than non-watermarked images. Meanwhile, the authors show that the proposed watermark is robust to various attacks and remains detectable.

**Questions:**

- I thought the watermark would be very vulnerable to a rotation attack since if you rotate the image, you mess up the bit sequence order. Do authors have any explanation for this? Or I might misunderstand something.
- Related to the above, would the watermark be robust to a flipping attack, such as horizontally flipping the image?

**Ethical Concerns:**

["NO or VERY MINOR ethics concerns only"]

**Final Justification:**

During rebuttal, the authors addressed most of my concerns by providing the additional experimental results. Therefore, I raised my score to 4.

**Limitations:**

Yes.

**Paper Formatting Concerns:**

No.

**Quality:**

2

**Strengths And Weaknesses:**

Strengths:
- The paper is well-written. The motivation is very strong.
- The method is simple, and it requires minimal effort for the model owner to deploy it.
- I really appreciate that the authors included various attacks, including one adaptive attack, in the paper.

Weaknesses:
- The contribution of this paper might be a little weak for me, as the authors simply apply the previous language model watermark to the Infinity model.
- The authors only conduct the experiments with the Infinity model. I think the paper will be strengthened if the authors can include more models. I know there might not be other bitwise autoregressive image models, but maybe apply a similar technique to bitwise language models?
- There is no comparison to other watermark techniques. I think the authors should at least compare to some post-hoc watermark methods.
- I wonder if the watermark is very radioactive, is a good thing? If so, it means the watermark can be easily learned by an adversary, and they can perform spoofing attacks easily. Ideally, we might want an undetectable watermark.

---

> ### Author Rebuttal · Authors · 2025-07-31
>
> We thank the Reviewer for the positive and constructive feedback. We appreciate that the Reviewer finds our paper well-written and the *motivation strong*. We are glad that our *comprehensive experiments* clearly demonstrate the *practicality* of our watermark and *robustness* to many attacks, including adaptive ones.
>
> In summary, our **BitMark generalizes to three additional models**, we **evaluated BitMark against post-processing watermarks**, extended the **robustness evaluation**, and **addressed concerns regarding radioactivity**. We provide case-by-case responses below:
>
> >**The contribution of this paper might be a little weak for me, as the authors simply apply the previous language model watermark to the Infinity model.**
>
> Recent watermarking papers [1,2] focus on the token level, where the token overlap between generated and re-encoded images is small. This necessitates retraining of the encoder and decoder. Our key contribution is to demonstrate that operating at the lower bit level offers a great advantage. Specifically, we show that the bit-level overlap between generated and re-encoded images is significant and sufficient to enable direct bit-level watermarking, without retraining the encoder and decoder, and this allows us to provide the strongest known watermark for image autoregressive models. As stated by Reviewer 9clc: *“the green-red list mechanism adapted from LLM watermarking is **cleverly** adapted to image generation scenario, with modifications to handle bitwise autoregression.”*
>
> >**The authors only conduct the experiments with the Infinity model. I think the paper will be strengthened if the authors can include more models.**
>
> We ran additional experiments to demonstrate that BitMark is applicable to Infinity-8B, as well as to two other bitwise models, namely BiGR [3] and Instella [4].  *BiGR* is an image generative model that harnesses binary latent codes and *Instella* is another text-to-image (T2I) Image AutoRegressive (IAR) model leveraging bitwise next-patch prediction.
>
> We extended our experimental analysis to include the additional models and added the experiments to Appendix E. We also report the results below:
>
> We calculated a representative p-value for 5000 watermarked images by first calculating the respective mean z-score, then standardizing this mean z-score using the mean and standard deviation derived from z-scores of 5000 MS-COCO images. $\Delta$ denotes the image quality difference to non-watermarked generated images. FID, KID & CLIP are reported for 10000 images.
>
> | Model  | p-value of mean z-score | $\Delta$ FID | $\Delta$ KID | $\Delta$ CLIP |
> |:-:|:-:|:-:|:-:|:-:|
> | Infinity 2B, $\delta=2$ | 1.0e-594 | -2.1169 | -0.0016 | 0.1086 |
> | Infinity 8B, $\delta=1.5$ | 1.3e-273 | 0.5299 | -0.0018 | 0.2792 |
> | BiGR 256x256, d24, $\delta=3$ | 1.16e-23 | -3.0304 | -0.0020 | 0.0017 |
> | BiGR 512x512, d32, $\delta=3$ | 3.37e-31 | -3.6660 | -0.0024 | 0.1329 |
> | Instella IAR, $\delta=1.5$ | 2.27e-69 | 1.1471 | 0.0004 | -0.8361 |
>
> The results show that we can effectively embed our BitMark into the images generated from Infinity 8B, BiGR, and Instella, without harm to the quality of output images (relatively small changes of the FID, KID, and CLIP scores). The extremely low p-values for the respective models demonstrate that we can confidently reject the $H_0$ (null hypothesis) and claim that our watermark was indeed embedded in the generated images. Thus, BitMark generalizes to multiple models and architectures.
>
> >**I know there might not be other bitwise autoregressive image models, but maybe apply a similar technique to bitwise language models?**
>
> Note that the LLMs do not suffer from the lack of token overlap, which is the case for Image AutoRegressive models (IARs). Our main observation is that the bit-wise overlap is significant in bit-wise IARs and enables us to embed the strong watermarks. Thus, bit-wise watermarking is not necessary for LLMs and the performance on the level of our BitMark can be already achieved on the token level. If the Reviewer can recommend any language models that utilize bit-wise generation, we would be happy to try our BitMark technique on them.
>
> >**There is no comparison to other watermark techniques. I think the authors should at least compare to some post-hoc watermark methods.**
>
> We added a comparison of BitMark ($\delta = 2$) against three postprocessing watermarks, namely RivaGAN [4] (with message length = 32), StegaStamp [5] (with message length = 100) and TrustMark [6] (with message length = 100). We applied the postprocessing watermarks on 1000 images generated by Infinity-2B and applied the same attacks from our paper. In the following we report the TPR@1%FPR. The experiments show that BitMark is the most robust watermark for Infinity (compared to RivaGAN, StegaStamp, and TrustMark).
>
> | Watermark vs. Attack | Average robustness | None | Noise | Blur | Color | Rotate | Vertical Flip | Horizontal Flip | Crop | JPEG | VAE | CtrlRegen |
> |:-:|:-:|:-:|:-:|:-:|:-:|:-:|:-:|:-:|:-:|:-:|:-:|:-:|
> | RivaGAN [5] | 72.1% | 99.7% | 98.3% | 99.7% | 99.4% | 96.7%  | 0% | 0% | 99.4% | 99.7% | 98.5% | 1.6% |
> | StegaStamp [6] | 64.6% | 100%  | 100%  | 100%  | 98.7% | 32.1%  | 1%            | 33.8% | 1% | 100% | 100% | 44.2% |
> | TrustMark [7] | 55.2% | 99.9% | 99.5% | 99.9% | 2.2% | 2.7% | 0.7% | 99.8% | 1.6% | 99.9% | 99.7% | 1.1% |
> | BitMark ($\delta = 2$) | 89.9% | 100% | 99.6% | 99.9% | 99.8% | 20.1% | 78.8% | 100% | 98.8% | 100% | 100% | 91.6% |
>
> >**​​If the watermark is very radioactive, is it a good thing? If so, it means the watermark can be easily learned by an adversary, and they can perform spoofing attacks. We want an undetectable watermark**
>
> Our primary motivation is to prevent model collapse since we want to ensure that companies like OpenAI do not use their own generated data or **any derivatives of their generated data** to train their future models. This is precisely where the radioactivity property becomes essential: the images generated by a model fine-tuned or trained on generated data are also harmful to the next models. Thus, the spoofing attack is not a concern in this case.
>
> >**I thought the watermark would be very vulnerable to a rotation attack since if you rotate the image, you mess up the bit sequence order.**
>
> The bit-sequence order in the latent space after tokenization is not directly transferable to the pixel space. While BitMark is applied in sequence order in the bit-space, due to the BSQ and Decoder, there are additional spatial transformations applied to the final model output. The important property that contributes to the robustness of BitMark is the large number of bits that are influenced during our watermark embedding.
>
> We performed an experiment computing the overlap between bits from: (1) the original generated image and the rotated encoded image, and (2) the encoded image and the rotated encoded image. Given that over 300k bits that are influenced by BitMark, even a small overlap of 50.14% at 20° is sufficient for the robust detection of our watermark.
>
>
> | **Rotation Degree** | 0° | 5° | 10° | 15° | 20° | 25° | 45° |
> |:-:|:-:|:-:|:-:|:-:|:-:|:-:|:-:|
> | **Average Bit Overlap** Between Generated and Rotated | 68.83% | 51.3%  | 50.37% | 50.22% | 50.14% | 50.09% | 50.05% |
> | **Average Bit Overlap** Between Encoded and Rotated | 100% | 51.66% | 50.46% | 50.26% | 50.18% | 50.12% | 50.06% |
>
>
> >**Related to the above, would the watermark be robust to a flipping attack, such as horizontally flipping the image?**
>
> We added an additional attack to our robustness evaluation and also reported the TPR@1%FPR in the following Table:
>
> | | horizontal flip | vertical flip |
> |:-:|:-:|:-:|
> | TPR@1%FPR | 100% | 78.8% |
>
> We observe that our BitMark is robust to the flipping attacks.
>
> ---
> If the above responses address some of the Reviewer's concerns, we would greatly appreciate it if they could consider updating their rating to reflect this.
>
> **References:**
>
> [1] Tong et al. “Training-Free Watermarking for Autoregressive Image Generation” https://arxiv.org/abs/2505.14673
>
> [2]  Jovanović et al. “Watermarking Autoregressive Image Generation” https://arxiv.org/abs/2506.16349
>
> [3] Hao et al. “BiGR: Harnessing Binary Latent Codes for Image Generation and Improved Visual Representation Capabilities” ICLR 2025. https://openreview.net/forum?id=1Z6PSw7OL8
>
> [4] Wang et al. “Instella-T2I: Pushing the Limits of 1D Discrete Latent Space Image Generation” https://arxiv.org/pdf/2506.21022
>
> [5] Kevin Alex Zhang, Lei Xu, Alfredo Cuesta-Infante, and Kalyan Veeramachaneni. Robust invisible video watermarking with attention. 2019.
>
> [6] Matthew Tancik, Ben Mildenhall, and Ren Ng. Stegastamp: Invisible hyperlinks in physical photographs. In IEEE/CVF Conference on Computer Vision and Pattern Recognition (CVPR), June 2020.
>
> [7] Tu Bui, Shruti Agarwal, and John Collomosse. Trustmark: Universal watermarking for arbitrary resolution images. ArXiv e-prints, November 2023.

---

> ### Author Response · Authors · 2025-08-04
> **Thank you for the acknowledgement and further clarification of the bit-sequence order**
>
> Dear Reviewer cNxC,
>
> We thank the Reviewer for acknowledging our rebuttal and are happy to address any concerns that still exist!
>
> In addition, we would like to further clarify the difference between token sequence and bit sequence as follows:
>
> An image can be tokenized into a $n \times n$ token matrix. Each token in this matrix has $m$ bits, where each bit is like as a *channel* of the matrix. Our watermarking method operates on the bit sequence across the $m$ channels (in each matrix token), not on the spatial arrangement of the matrix tokens.
>
> For example, when we apply horizontal flipping to the matrix, we change the order of the columns in the $n \times n$ matrix. However, the order of the $m$-bit sequence in each token stays almost the same. Since the watermark is embedded in the $m$-bit sequences rather than the spatial relationships between the $n \times n$ token matrix, horizontal flipping largely preserves the watermark signal.
>
> We also would like to follow up on our answers, especially regarding our contribution, our application of BitMark to other models, comparison to other watermark techniques, and the radioactivity. Do our replies adequately address the reviewer's concerns? We are happy to provide any additional explanations. If our answers address the Reviewer's questions, we hope that the reviewer will consider raising the rating for our submission.
>
> With kind regards,
>
> Authors of paper #2950 (BitMark)

---

> > ### Comment · Reviewer_cNxC · 2025-08-05
> >
> > Thanks for your detailed response. I think additional results do show the robustness of the proposed method. Therefore, I have raised my score.

---

> > > ### Author Response · Authors · 2025-08-06
> > > **Thank you!**
> > >
> > > Dear Reviewer cNxC,
> > >
> > > We thank the Reviewer for the engagement in the discussion and are happy that our detailed responses addressed the concerns. We also appreciate your feedback since the paper improved as a result. We are grateful for increasing the score.
> > >
> > > With kind regards,
> > >
> > > Authors of paper #2950 (BitMark)

---

### Official Review · Reviewer_7q5w · 2025-06-30

**Clarity:** 4
**Significance:** 3
**Originality:** 2
**Rating:** 4
**Confidence:** 3

**Summary:**

The paper introduces BitMark, a novel bitwise watermarking method for the Infinity image generation model. BitMark embeds imperceptible signals at the bit level during image generation, allowing detection of generated content while preserving image quality and speed. BitMark is the first watermarking scheme for Infinity, with some robustness and radioactivity.

**Questions:**

1.  Results on Infinity-8B will be beneficial for this paper's generalization ability.
2.  The Radioactive experiment should be conducted with various watermarked ratios on training data.
3.  What is the superiority of this work compared with previous post-hoc watermarking schemes?

**Ethical Concerns:**

["NO or VERY MINOR ethics concerns only"]

**Final Justification:**

The rebuttal delivered far more results, which addressed most of my concerns. I would raise my ratings.

**Limitations:**

yes

**Quality:**

3

**Strengths And Weaknesses:**

Strengths:
+ The first watermarking scheme for Infinity.
+ Evaluated detectable ability and robustness.
+ Clear motivation description.

Weaknesses:
- Generalization ability among models should be further evaluated.
- Radioactivity is not evaluated enough.
- It would be better if compared with baselines.

The authors also successfully evaluated the effectiveness and robustness of this work on Infinity-2B.
The paper is well-written. Section 3.1 clearly explains the motivation of this work.

However, this work would also benefit from some improvements.
First, although this work evaluated its effectiveness on Infinity-2B and explained the reason for the absence of other models, only the Infinity-2B model is evaluated. Without generalization among different models, the real effectiveness of this work is undecided.

Second, as the main claimed strength of this work, the radioactivity is not evaluated enough. The authors' fine-tuned model on the dataset only consists of watermarked data. In practical training, we cannot assume that only watermarked images from the same model are used in fine-tuning.
The dataset should be evaluated, consisting of both watermarked and benign images, together with images generated from the model with different green-red settings.

Third, baselines should be compared with this work to demonstrate the superiority of this work, especially post-hoc baselines, which previous works [1,2] included to compare.

---

[1] Gunn, Sam, Xuandong Zhao, and Dawn Song. "An undetectable watermark for generative image models." The Thirteenth International Conference on Learning Representations.

[2] Wang, Ziyi, et al. "Safe-var: Safe visual autoregressive model for text-to-image generative watermarking." arXiv preprint arXiv:2503.11324 (2025).

---

> ### Author Rebuttal · Authors · 2025-07-31
>
> We thank the Reviewer for the positive and constructive feedback. We appreciate that the Reviewer finds our paper to be well written and clearly motivated.
>
> In summary, we **generalized BitMark** to three additional models, **evaluated BitMark against post-processing watermarks** and extended the **radioactivity experiments**. We address the points in detail below:
>
> >**Results on Infinity-8B will be beneficial for this paper's generalization ability. Generalization ability among models should be further evaluated.**
>
> We ran additional experiments to demonstrate that BitMark is applicable to Infinity-8B, as well as to two other bitwise models, namely BiGR [1] and Instella [2].  *BiGR* is an image generative model that harnesses binary latent codes and *Instella* is another text-to-image (T2I) Image AutoRegressive (IAR) model leveraging bitwise next-patch prediction.
>
> We extended our experimental analysis to include the additional models and added the experiments to Appendix E. We also report the results below:
>
> We calculated a representative p-value for 5000 watermarked images by first calculating the respective mean z-score, then standardizing this mean z-score using the mean and standard deviation derived from z-scores of 5000 MS-COCO images. $\Delta$ denotes the image quality difference to non-watermarked generated images. FID, KID & CLIP are reported for 10000 images.
>
> | Model  | p-value of mean z-score | $\Delta$ FID | $\Delta$ KID | $\Delta$ CLIP |
> |:-:|:-:|:-:|:-:|:-:|
> | Infinity 2B, $\delta=2$ | 1.0e-594 | -2.1169 | -0.0016 | 0.1086 |
> | Infinity 8B, $\delta=1.5$ | 1.3e-273 | 0.5299 | -0.0018 | 0.2792 |
> | BiGR 256x256, d24, $\delta=3$ | 1.16e-23 | -3.0304 | -0.0020 | 0.0017 |
> | BiGR 512x512, d32, $\delta=3$ | 3.37e-31 | -3.6660 | -0.0024 | 0.1329 |
> | Instella IAR, $\delta=1.5$ | 2.27e-69 | 1.1471 | 0.0004 | -0.8361 |
>
> The results show that we can effectively embed our BitMark into the images generated from Infinity 8B, BiGR, and Instella, without harm to the quality of output images (relatively small changes of the FID, KID, and CLIP scores). The extremely low p-values for the respective models demonstrate that we can confidently reject the $H_0$ (null hypothesis) and claim that our watermark was indeed embedded in the generated images. Thus, BitMark generalizes to multiple models and architectures.
>
>
> >**The Radioactive experiment should be conducted with various watermarked ratios on training data.**
>
> We performed additional experiments on the radioactivity with various ratios of watermarked data. The experiments clearly show that with an increasing amount of watermarked data, the signal of BitMark becomes stronger. Even if model M2 is trained on only 10% of our watermarked data, the model M2 can already generate images with a clear bias towards our watermark distribution, resulting in a high TPR of 22.7% @ 1% FPR.
> We note that even a slight bias is significant at the distribution level. Concretely, we apply a statistical test, namely the Mann-Whitney-U-Test, to test if the z-score distribution of M2-generated data is significantly larger than on M1 generated data. The test is significant with a  p-value of 5.6e-62 when M2 is trained on 5% watermarked data, showing that the z-score distribution of M2-generated data is greater than the and clean data z-scores.
>
> **As shown in the following table, less than 5% of watermarked data in the training set is enough to distinguish if Infinity-2B has been trained on watermarked data.**
>
> | Metric  \ p% of watermarked data | Clean model | Finetune (0) | 1     | 5       | 10       | 25    | 50   | 100  |
> |----------------------------------|-------------|-------------|-------|---------|----------|-------|------|------|
> | TPR@1%FPR                        | 0.1%        | 0.3%        | 1%    | 2.9%    | 22.7%    | 98.3% | 100% | 100% |
> | Mann-Whitney-U-Test (p value)    | 0.999       | 0.999       | 0.999 | 5.6e-62 | 6.8e-209 | 0     | 0    | 0    |
>
>
> >**Fine-tuning models on images generated with different green-red settings**
>
> In our setup, a model provider decides on a specific green-red setting instead of using many of them. This makes the watermark detection much easier for the model provider.
>
> >**What is the superiority of this work compared with previous post-hoc watermarking schemes?**
>
>
> We added a comparison of BitMark ($\delta = 2$) against three postprocessing watermarks, namely RivaGAN [3] (with message length = 32), StegaStamp [4] (with message length = 100) and TrustMark [5] (with message length = 100). We applied the postprocessing watermarks on 1000 images generated by Infinity-2B and applied the same attacks from our paper. In the following we report the TPR@1%FPR. The experiments show that BitMark is the most robust watermark for Infinity (compared to RivaGAN, StegaStamp, and TrustMark).
>
> | Watermark vs. Attack | Average robustness | None | Noise | Blur | Color | Rotate | Vertical Flip | Horizontal Flip | Crop | JPEG | VAE | CtrlRegen |
> |:-:|:-:|:-:|:-:|:-:|:-:|:-:|:-:|:-:|:-:|:-:|:-:|:-:|
> | RivaGAN [3] | 72.1% | 99.7% | 98.3% | 99.7% | 99.4% | 96.7%  | 0% | 0% | 99.4% | 99.7% | 98.5% | 1.6% |
> | StegaStamp [4] | 64.6% | 100%  | 100%  | 100%  | 98.7% | 32.1%  | 1%            | 33.8% | 1% | 100% | 100% | 44.2% |
> | TrustMark [5] | 55.2% | 99.9% | 99.5% | 99.9% | 2.2% | 2.7% | 0.7% | 99.8% | 1.6% | 99.9% | 99.7% | 1.1% |
> | BitMark ($\delta = 2$) | 89.9% | 100% | 99.6% | 99.9% | 99.8% | 20.1% | 78.8% | 100% | 98.8% | 100% | 100% | 91.6% |
>
> ---
> If the above responses address some of the Reviewer's concerns, we would greatly appreciate it if they could consider updating their rating to reflect this.
>
> **References:**
>
>
> [1] Hao et al. “BiGR: Harnessing Binary Latent Codes for Image Generation and Improved Visual Representation Capabilities” ICLR 2025.
>
> [2] Wang et al. “Instella-T2I: Pushing the Limits of 1D Discrete Latent Space Image Generation”. ArXiv e-prints, June 2025
>
> [3] Kevin Alex Zhang, Lei Xu, Alfredo Cuesta-Infante, and Kalyan Veeramachaneni. Robust invisible video watermarking with attention. 2019.
>
> [4] Matthew Tancik, Ben Mildenhall, and Ren Ng. Stegastamp: Invisible hyperlinks in physical photographs. In IEEE/CVF Conference on Computer Vision and Pattern Recognition (CVPR), June 2020.
>
> [5] Tu Bui, Shruti Agarwal, and John Collomosse. Trustmark: Universal watermarking for arbitrary resolution images. ArXiv e-prints, November 2023.

---

> > ### Comment · Reviewer_7q5w · 2025-08-03
> >
> > Thank you for detailed responses and new data. I have raised my ratings

---

> > > ### Author Response · Authors · 2025-08-04
> > > **Detailed responses and new data**
> > >
> > > Dear Reviewer 7q5w,
> > >
> > > We thank the Reviewer for the engagement in the discussion and are glad that our detailed responses and new data addressed the concerns.
> > >
> > > We appreciate the feedback you gave us and increasing the rating.
> > >
> > > With kind regards,
> > >
> > > Authors of paper #2950 (BitMark)

---

### Official Review · Reviewer_9cLc · 2025-07-01

**Clarity:** 3
**Significance:** 3
**Originality:** 3
**Rating:** 5
**Confidence:** 2

**Summary:**

This paper introduces BitMark, a novel watermarking method for bitwise AR model. It addresses the risk of model collapse training on AI-generated content. BitMark embeds watermarks at the bit level, leveraging the observation that bitwise discrepancies in autoregressive models are smaller than token-level ones. It uses a green-red list mechanism to subtly bias bit generation, preserving image quality and generation speed while enabling reliable detection.

**Questions:**

Can attackers reverse-engineer the green list by analyzing large sets of generated images? If so, how does BitMark defend against such attack?

**Ethical Concerns:**

["NO or VERY MINOR ethics concerns only"]

**Final Justification:**

Thanks authors for the reply. My concerns are addressed and I am keeping my score as accept.

**Limitations:**

yes

**Quality:**

3

**Strengths And Weaknesses:**

Strengths:
(1) The paper is well written and easy to follow.
(2) The design choice is well-motivated, for example choosing bitwise watermarking rather than token-wise watermarking. Also, the green-red list mechanism adapted from LLM watermarking is cleverly adapted to image generation scenario, with modifications to handle bitwise autoregression.
(3) The experiments are comprehensive, which shows robust performance on image quality and generation speed. Additionally, the detection process is efficient (~0.5 seconds per image), ensuring the method remains practical for real-world deployment.

Weaknesses:
(1) The method struggles with large rotation attacks. While the paper suggests that rotation estimation tools could rotate back and correct these images (, thus being easy to be detected), no experimental evidence supports this claim.

---

> ### Author Rebuttal · Authors · 2025-07-31
>
> We thank the Reviewer for the positive and constructive feedback. We appreciate that our work is considered *novel* and that the green-red list mechanism adapted from LLM watermarking to image generation is described as *clever*. We are glad that our *comprehensive experiments* clearly demonstrate the *practicality* of our watermark.
>
> In summary we **implemented the rotation estimation tool** to revert rotations and observed that the detection of our watermark can be increased with the rotation correction tools. We also addressed the question regarding **reverse engineering the green list**, in essence: the choice of the arbitrary length of BitMark makes a frequency analysis (used to reverse-engineer the green list) difficult.
>
> We address the points in detail below:
>
> >**Weaknesses: (1) The method struggles with large rotation attacks. While the paper suggests that rotation estimation tools could rotate back and correct these images (thus being easy to be detected), no experimental evidence supports this claim.**
>
> We performed experiments to detect BitMark after correcting rotated images with the rotation estimation tool referred to in our submission [1]. Specifically, we first applied random rotations to the watermarked images from 0 to 360 degrees. We then used the rotation estimation tool [1] to estimate the rotation angles of the images and rotated them back. Our BitMark is then detected on the re-rotated images. The following table showing the detection results with and without the rotation correction was added to Appendix E.
>
> | Setting | AUC (%) | Accuracy (%) | TPR@1%FPR (%) |
> |:-:|:-:|:-:|:-:|
> | No rotation correction | 60.4 | 60.8 | 18.4 |
> | After  rotation correction | 99.5 | 99.2 | 98.3 |
>
> The results show that BitMark achieves a significant AUC of 99.5% and a TPR of 98.3% at 1% FPR after the rotation correction. This demonstrates that the detection of our watermark can be increased with the rotation correction tools.
>
> In addition, we consider an even harder case: after the random rotation, we apply a central crop to the image with 50% cropping ratio, such that all the black corners during rotation are completely removed. We show the results below (also added them to Appendix E)..
>
> | Setting | AUC (%) | Accuracy (%) | TPR@1%FPR (%) |
> |:-:|:-:|:-:|:-:|
> | No rotation correction | 59.8 | 58.9 | 9.9 |
> | After  rotation correction | 89.1 | 86.5 | 66.3 |
>
> Even in this harder case, the detection of our watermark is also increased after the rotation correction. Notably, our BitMark achieves a high AUC of 89.1 and a TPR of 66.3 at 1% FPR.
>
> >**Can attackers reverse-engineer the green list by analyzing large sets of generated images? If so, how does BitMark defend against such an attack?**
>
> The attacker cannot reverse the green list since, in our watermark, we follow the Private Watermarking from KGW [13], where the generative model is kept behind a secure API. The green-list can only be reverse-engineered if the attacker could re-encode and tokenize the images. The choice of the arbitrary length of BitMark makes a frequency analysis (used to reverse-engineer the green list) difficult, as the frequency analysis has to be done for multiple possible patterns of size n. Additionally, even the knowledge about the green list is not effective in removing our watermark, which we show with our Bit-Flipper attack. The attack cannot remove our BitMark without causing a severe image degradation.
>
> ---
> We would like to thank the Reviewer again for the questions and comments. The paper has definitely improved as a result. We would like to check if there are any other questions or concerns that we can address.
>
> **References:**
>
> [1] Maji Subhadip and Smarajit Bose. "Deep image orientation angle detection." 2020.
>
> [2] John Kirchenbauer et al. “A watermark for large language models.” ICML, 2023.

---

> > ### Comment · Reviewer_9cLc · 2025-08-05
> >
> > Thanks authors for the reply. My concerns are addressed and I am keeping my score as accept.

---

### Author Response · Authors · 2025-08-08
**Thank you!**

Dear Area Chair and Reviewers,

Thank you for your time and effort in evaluating our work. Throughout the rebuttal phase, we carefully addressed all Reviewers’ comments, particularly those requesting additional experiments. In response, we conducted new experiments, presented detailed results, and sought to clarify any misunderstandings that have arisen from the initial reviews.

To summarize, BitMark is a **“novel watermarking method”** (9clc, 7q5w) which “cleverly adapts the red-green list mechanism” (9cLc). Our proposed threat model has been appreciated as a clear and **strong motivation** (cNxC, 7q5w) as well as real and increasingly urgent, important and timely (NhKn). The Reviewers valued the comprehensive experiments (9cLc), acknowledging the various attacks, including conventional, advanced, as well as an adapted white-box attack (NhKn, cNxC). Our rebuttal has been appreciated as extensive (7q5w, NhKn) and convinced all Reviewers with a borderline-score to raise their rating (7q5w, cNxC, NhKn). During the discussion, further uncertainties have been addressed (cNxC). The feedback after the discussion is unanimously positive.

Once again, we are grateful for your constructive feedback and consideration. Please do not hesitate to let us know if there is anything further we can clarify or address at this stage.

With kind regards,

Authors

---

### Public Comment · ~Qiran_Lai1 · 2025-11-11
**Can you provide your GitHub code?**

Your paper have codes Link but it cannot work. Can you provide it on the Github again? Thank you very much.

---

> ### Public Comment · ~Adam_Dziedzic1 · 2026-01-13
> **Link to the GitHub code**
>
> Hi Qiran Lai,
>
> This is the link to our GitHub code: https://github.com/sprintml/BitMark
>
> With kind regards,
>
> Authors

---

### Decision · Program_Chairs · 2025-09-17

**Decision:**

Accept (poster)

**Comment:**

This paper proposes a watermarking framework for bitwise autoregressive image generative models. It adapts the green-red list mechanism from LLM watermarking to image models and embeds a watermark at the bit level of the token stream during inference.

It is the first watermarking method tailored for bitwise AR image generation. Reviewers appreciate its clear motivation and strong robustness against many attack types. It also demonstrates radioactivity across models that enables provenance tracking. Experiments show negligible image quality impact.

Reviewers had concerns initially on its generalization ability on different models and datasets, as well as the sufficiency of experimental results on its robustness against attacks and radioactivity. Authors' rebuttal provided additional experiments and clarification that addressed most concerns. As a result, reviewers reached a consensus of acceptance.

Some remaining concerns, though not critical for acceptance, are:
- Moderate novelty as this method is an adaptation of existing LLM watermarking method to bitwise image generation
- The mechanism under its radioactivity is still not very theoretically well-justified
- Although it compares with several baselines, it lacks high-level comparison and justification of where the superiority of this work is from compared with previous post-hoc watermarking schemes